# Game theoretical inference of human behavior in social networks

Nicolò Pagan [1]* & Florian Dörfler [1]

Social networks emerge as a result of actors' linking decisions. We propose a game-theoretical model of socio-strategic network formation on directed weighted graphs, in which every actors' benefit is a parametric trade-off between centrality measure, brokerage opportunities, clustering coefficient, and sociological network patterns. We use two different stability definitions to infer individual behavior of homogeneous, rational agents from network structure, and to quantify the impact of cooperation. Our theoretical analysis confirms results known for specific network motifs studied previously in isolation, yet enables us to precisely quantify the trade-offs in the space of user preferences. To deal with complex networks of heterogeneous and irrational actors, we construct a statistical behavior estimation method using Nash equilibrium conditions. We provide evidence that our results are consistent with empirical, historical, and sociological observations on real-world data-sets. Furthermore, our method offers sociological and strategic interpretations of random networks models, such as preferential attachment and small-world networks.

[1] Automatic Control Laboratory, ETH Zürich, Physikstrasse 3, 8092 Zürich, Switzerland. *email: pagann@control.ee.ethz.ch

There has been a growing interdisciplinary interest in the study of social networks over the past few decades. Especially since the diffusion of on-line platforms, e.g., Twitter or Instagram, there is evidence that both sociological and strategic behavior play an important role[1]. Thus, it became important to understand how networks are formed[2] and especially how networks affect actors' behavior and vice versa.

Starting from the random graph model proposed by Erdös and Rényi[3,4], the complex networks community developed a number of network formation models driven by sociological observations and supported by empirical evidence. Among them, the small-world network model introduced by Watts and Strogatz[5] shows that the addition of few random ties to a regular lattice (highly locally connected) results into a small diameter network, as in Milgram's experiment[6] on the six degrees of separation. To explain the emergence of scaling in random networks, Barábasi and Albert proposed the preferential attachment model[7], in which newborn nodes select their connections proportional to popularity. A broad literature on complex (social) networks and dynamics thereof has grown ever since (see refs. [8–10]). While such probabilistic models can successfully reproduce the macroscopic statistical structural properties of social networks, they do not offer insights into the sociological microscopic foundations.

One such socio-theoretical and statistical approach was proposed by Snijders[11] with the Stochastic Actor-Oriented Models (SAOM). They consider observed networks as the result of the actors' linking behavior[12] assuming that social actors can change their outgoing ties. The payoff function that each actor tries to maximize is split into a modeled and a random component, with the former containing statistical parameters that can be estimated from available data through likelihood-based methods[13]. The modeled component is assumed to be a linear combination of effects, e.g., reciprocity, transitivity, or the tendency of having ties at all[13]. Similarly to SAOM, Exponential Random Graph Models (ERGM)[14] study network configurations, which are small subsets of possible network ties (and/or actors' attributes), e.g., reciprocated ties[15]. Yet, the focus is on ties rather than on actors.

In the economics community, a plausible and widely supported belief is that actors strategically choose their relations to optimize their network positions in an incentive-guided fashion[16]. Similarly to SAOM, strategic network formation models assume that actors aim at maximizing payoff functions that depend on their position in the network and on the topology. The objective is to explain why certain network architectures emerge when actors strive for centrality, while links are costly. The literature on this topic is broad[17–27] (see refs. [28–31] for extensive surveys), yet there is no common agreement on the specific centrality metrics[32]. Among the seminal works on strategic network formation, Bala and Goyal[18] use degree centrality, while the connections model introduced by Jackson and Wolinsky[17] is related to closeness centrality, as shown in ref. [22]. Others[19–21] propose models where actors strive for structural holes, which are missing connections between certain pairs of agents, thus brokerage opportunities. Burt showed that his brokerage constraint measure, defined in ref. [33], is tightly related to betweenness centrality[34]. According to Coleman[35], triangulated structures provide cohesive support to the agents. Davis[36] also showed empirically that transitivity, often termed network (or triadic) closure or clustering[15,37], is a prevalent effect in many human social networks as the result of social selection based on, e.g., homophily[38].

Depending on the choice of payoff function, different models evince a relation between centrality metrics and the stability of specific network architectures. For instance, refs. [20,21] show that when actors strive for betweenness centrality, balanced complete bipartite (or more generally multipartite) networks are stable. Conversely, closeness centrality incentives lead to star-like

architectures[22], and complete networks are stable when closed triads are beneficial[23]. The main limitation of these models lies in the isolation of specific centrality metrics and network motifs, which prevents from a comprehensive analysis of the network topology stability with respect to multiple co-existing incentives. Some preliminary attempts in overcoming this limitation can be found in ref. [23], where the authors compare the approaches based on Coleman and Burt's theories and experimentally show that the spectrum of stable networks effectively depends on the trade-off between social support and brokerage; and in ref. [24] where betweennes and closeness centralities are simultaneously investigated.

Besides adopting different payoff functions, the literature on strategic network formation exhibits a variety of assumptions: most works consider unweighted and undirected networks, though some variations have been studied (see ref. [25] for a directed version of the connections model, and see ref. [26] for weighted graphs). Yet, while some models rely on agents having perfect information of the network, an active line of research investigates more realistic incomplete information scenarios, e.g., ref. [39].

In this work, we propose a socio-strategic network formation model whose payoff function is characterized by a parametric combination of a locally assessable Katz centrality[40] and a clustering coefficient. This allows us to represent a wide range of strategic actors' incentives, from indegree to closeness centrality, or from clustering coefficients to betweenness-type centrality. Moreover, we capture the significant difference between follower and followee using directed networks. Despite having received little attention in the strategic network formation literature, they allow us to emphasize the socio-theoretical interpretation of our payoff function in terms of the effects studied in SAOM.

Our first objective is to theoretically study the relation between homogeneous actors' behavior and stable network motifs. We do so by means of two different notions of stability: Nash and pairwise-Nash equilibrium. With our parametric model and our analytical proof methods we not only confirm phenomena previously observed in isolation, but we also provide a comprehensive quantitative analysis and we discover transition paradigms in the space of individuals' preferences.

Secondly, by means of the socio-theoretical interpretation of our payoff function, we propose a statistical method, based on Nash equilibrium conditions. The method enables us to perform individual behavior estimation on real-world networks of heterogeneous and not necessarily rational agents. We validate our predictions on the Medici's strategic behavior in Renaissance Florence[41] and on the hierarchical network of confiding relationships within an Australian bank[42].

Third and finally, we show that our behavior estimation method can shed light on a sociological and strategic interpretation of complex random networks models.

## Results

**Network formation problem setup.** Let $\mathcal{N} = \{1, \dots, N\}$ be a finite set of actors or agents (with $N \geq 3$ to avoid trivial cases). Depending on the application, an actor may be a human being, a firm, a country, or some other autonomous entity. Agents are endowed with a payoff function, and they are assumed to be rational, thus they aim at maximizing it. They do so in a myopic fashion, i.e., without anticipating others' potential reaction. The network relations among these agents are formally represented by weighted and directed graphs $\mathcal{G}$ without self-loops, whose nodes are identified with the set $\mathcal{N}$ of agents and whose arcs weights $a_{ij}$ and $a_{ji}$, in the range between 0 and 1, denote the strength of the directed relations among agents $i$ and $j$.

We assume that each agent of the network has control on the weights of her outgoing links, while she cannot affect her incoming links. In other words, she can decide her followees but not her followers. In game-theoretical language, a typical action of agent $i$ can be expressed by

$$\mathbf{a}_i = \left[ a_{i1}, \ldots, a_{i,i-1}, 0, a_{i,i+1}, \ldots, a_{iN} \right],$$

living in the action space $\mathcal{A} = [0, 1]^{N-1}$. Conversely, we denote a typical action of all agents but $i$ as

$$\mathbf{a}_{-i} = \left[ \mathbf{a}_1; \ldots; \mathbf{a}_{i-1}; \mathbf{a}_{i+1}; \ldots; \mathbf{a}_N \right] \in \mathcal{A}^{N-1}.$$

**Payoff function.** In a directed network setting, perhaps the simplest measure of centrality is the indegree centrality (e.g., number of followers), which is based on the sum of incoming ties (see Fig. 1a). Formally, the indegree centrality of agent $i$ is measured as $\sum_k a_{ki}$. In the connections model by Jackson and Wolinksy[17] benefits also come from indirect connections, given by $\delta^t v$, where $t$ is the length of the shortest path connecting two agents, $v$ is a fixed parameter, and $\delta \in (0, 1)$ is a decay factor. Similarly, we use $\delta_i \in [0, 1]$ and define

$$t_i(\mathbf{a}_i, \mathbf{a}_{-i}, \delta_i) = \sum_k a_{ki} + \delta_i \sum_l \sum_k a_{lk} a_{ki} + \delta_i^2 \sum_m \sum_l \sum_k a_{ml} a_{lk} a_{ki}$$

as a measure of the influence of agent $i$ in the network. Such a measure extends the indegree centrality definition (which can be recovered by setting $\delta_i = 0$) by introducing the contribution of the strength of all weighted paths of length 2 and 3 which are ending in $i$, discounted with factors $\delta_i$ and $\delta_i^2$, as shown in Fig. 1b. This measure can also be viewed as an approximated Katz centrality. In the original definition, Katz[40] considers paths of all lengths, yet in real-world social networks agents have limited information on the network topology (one can think of Linkedin's 3rd degree of separation). Compared to that, our definition is locally assessable, i.e., it does not require complete information of the entire network, yet it includes most important social networks patterns, such as diads and triads[15].

As discussed in the introduction, agents may privilege social support. Formalized in the network settings, agents benefit from being surrounded by closed triads, or in other words when a friend of a friend is a friend[43]. In graph theory, the mean probability that two nodes, which are network neighbors of the same other node, will themselves be neighbors is referred as clustering coefficient[5]. Albeit it might be hard for agents to compute such a probability, they can estimate, for each friend $k$, the number of common friends $l$. Similarly to the approach in ref. [23], we define the clustering of agent $i$ as

$$u_i(\mathbf{a}_i, \mathbf{a}_{-i}) = \sum_k a_{ik} \left( \sum_l a_{il} a_{lk} \right),$$

as depicted in Fig. 1c, meaning that the friendship from $i$ to $k$ is more valuable the higher the number of common friends $l$ between $i$ and $k$. Finally, as typically done in the strategic network formation literature, e.g., refs. [17,21,24], we model the cost to agent $i$ as the effort required for agent $i$ to maintain the link towards the other agents $k$, thus proportionally to agent $i$'s outgoing ties $a_{ik}$. The cost, illustrated in Fig. 1d, results in

$$c_i(\mathbf{a}_i) = \sum_k a_{ik}.$$

Alternatively, as a minor variation of the model, a quadratic cost function as in ref. [23] can be used to model ties with non-constant marginal costs reflecting the fact agents have to divide their attention over all their relationships.

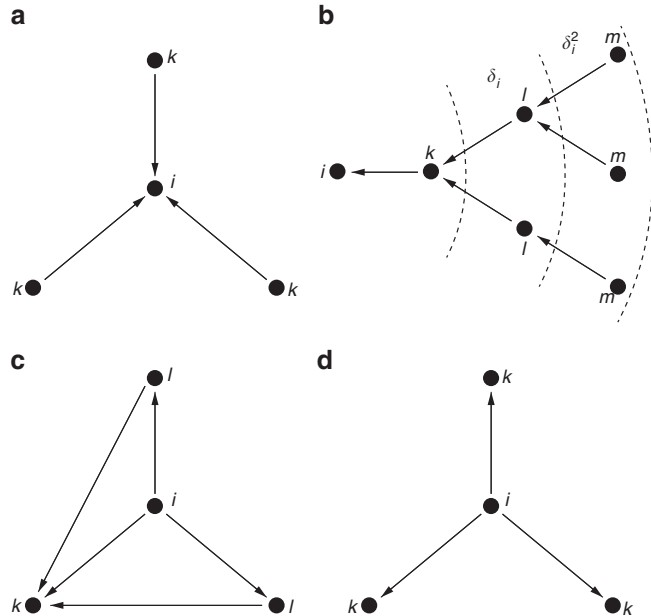

**Fig. 1** Sketch of the payoff function's contributions. **a** The indegree centrality of agent $i$ is determined by the sum of all the incoming ties. **b** The influence of agent $i$ is computed summing up the weighted paths to the node $i$, discounted by 1, $\delta_i$, $\delta_i^2$. **c** The clustering coefficient of agent $i$ depends on the weighted directed closed triads that surround node $i$. **d** The cost to agent $i$ corresponds to the outdegree of node $i$.

In summary, we assume every agent $i \in \mathcal{N}$ is endowed with a parametric payoff function $V_i$ which depends on $a_i$, the action of agent $i$, and on $\mathbf{a}_{-i}$, the action of all other agents:

$$V_i(\mathbf{a}_i, \mathbf{a}_{-i}, P_i) = \alpha_i t_i(\mathbf{a}_i, \mathbf{a}_{-i}, \delta_i) + \beta_i u_i(\mathbf{a}_i, \mathbf{a}_{-i}) - \gamma_i c_i(\mathbf{a}_i).$$

The individual set of parameters $P_i = \{ \alpha_i, \beta_i, \gamma_i, \delta_i \}$ is composed by real numbers, with $\alpha_i \geq 0$, $\gamma_i > 0$, and $\delta_i \in [0, 1]$, which allow payoff tuning according to the actor's preferences. For instance, large $\alpha_i$ and $\delta_i$ make the influence measure more valuable, while higher values of $\gamma_i$ increase the cost of maintaining links. Note that $\beta_i$ can take negative values. In this case, the clustering coefficient $u_i$ acts as a cost. Drawing inspiration from ref. [23], this enables us to measure the absence of direct brokerage opportunities and to model a number of contexts in which agents prefer ties with unconnected others, as in Burt's theory of structural holes[33]. Albeit this cost does not correspond to the original constraint measure constructed by Burt, it preserves the underlying intuition that agents are more constrained by their network if they have many redundant contacts. Hence, negative values of $\beta_i$ are aligned with Burt's theory, and thus with betweenness centrality, whereas positive values support transitivity and network closure, according to Coleman's theory.

**Socio-theoretical interpretation.** Although our payoff function emerges as a parametric generalization of previous models in the network formation literature, a link to network dynamics models from social science theory, e.g., SAOM, can be established. If we focus on the extended indegree centrality measure $t_i(\mathbf{a}_i, \mathbf{a}_{-i}, \delta_i)$, by isolating agent $i$'s contribution we obtain the following expression (the derivation can be found in the proof of Theorem

3 in Supplementary Note 1)

$$t_i(\mathbf{a}_i, \mathbf{a}_{-i}, \delta_i) = f_i(\mathbf{a}_{-i}, \delta_i) + \delta_i \underbrace{\left( \sum_{k \neq i} a_{ik} a_{ki} \right)}_{\text{rec}(\mathbf{a}_i, \mathbf{a}_{-i})}$$

$$+ \delta_i^2 \left( \underbrace{\left( \sum_{k \neq i} a_{ik} a_{ki} \right) \underbrace{\sum_{m \neq i} a_{mi}}_{\text{indeg}(\mathbf{a}_{-i})}}_{\text{rec}(\mathbf{a}_i, \mathbf{a}_{-i})} + \underbrace{\left( \sum_{l \neq i} a_{il} \sum_{k \neq i,l} a_{lk} a_{ki} \right)}_{\text{cycles}(\mathbf{a}_i, \mathbf{a}_{-i})} \right),$$

where $f_i(\mathbf{a}_{-i}, \delta_i)$ denotes the contribution which does not depend on $i$'s action. In other words, the extended indegree centrality measure includes, among others, reciprocal and three-cycle structures, denoted, respectively, as $\text{rec}(\mathbf{a}_i, \mathbf{a}_{-i})$ and $\text{cycles}(\mathbf{a}_i, \mathbf{a}_{-i})$, and sketched in Fig. 2a, b.

Thus, the payoff function can be conveniently re-written in the following alternative formulation:

$$\begin{aligned} V_i(\mathbf{a}_i, \mathbf{a}_{-i}, P_i) = & \alpha_i f_i(\mathbf{a}_{-i}, \delta_i) + \alpha_i \delta_i \text{rec}(\mathbf{a}_i, \mathbf{a}_{-i}) \\ & + \alpha_i \delta_i^2 (\text{rec}(\mathbf{a}_i, \mathbf{a}_{-i}) \text{indeg}(\mathbf{a}_{-i}) + \text{cycles}(\mathbf{a}_i, \mathbf{a}_{-i})) \\ & + \beta_i u_i(\mathbf{a}_i, \mathbf{a}_{-i}) - \gamma_i c_i(\mathbf{a}_i), \end{aligned}$$

where the dependency on basic sociological effects (as in SAOM) or configurations (ERGM) has been highlighted.

**Stability.** In game-theoretic modeling of strategic network formation, two stability notions are typically considered: Nash equilibrium and pairwise stability. The former is based on the idea that agents act purely selfishly: the network is stable when no agent can be better off by unilaterally deviating from her equilibrium strategy. In the latter, a network is stable when no pair of agents can coordinate in order to be both better off. Both notions have been extensively used in strategic network formation analysis. Among the pioneering works, Jackson and Wolinsky[17] and Bala and Goyal[18] used, respectively, pairwise stability and Nash equilibrium.

Following these two classes, we define two types of stability. First, we model purely selfish actors through the Nash equilibrium notion. This is a reasonable approach in many competitive contexts or marketing environments, e.g., when agents strategically retweet or choose their Instagram followees. The formal definition is as follows.

**Definition.** $\mathcal{G}^\star$ is a Nash equilibrium (NE) if
**C1.** *for all agents $i$, $V_i(\mathbf{a}_i, \mathbf{a}^\star_{-i}) \leq V_i(\mathbf{a}^\star_i, \mathbf{a}^\star_{-i})$, $\forall \mathbf{a}_i \in \mathcal{A}$.*

Note that agents are allowed to play any action in the space $\mathcal{A}$, i.e., to simultaneously change all the outgoing ties.

Second, we propose the pairwise-Nash stability definition, which combines the selfish attitude with the possibility of coordination among agents, and thus is more suitable to model situations where actors are open to cooperation, while also being naturally selfish. Indeed, in many social and economic networks, it is not uncommon to observe cooperation, and previous works (see refs. [44,45]) already made use of pairwise-Nash equilibrium, also referred as Bilateral equilibrium (see ref. [19]). Practically, agents are allowed to deviate alone or by pairs. Thus, instead of considering every agent singularly, we consider meetings. In a meeting between $i$ and $j$, agent $i$ can only revise the tie $a_{ij}$, while, simultaneously, $j$ can modify $a_{ji}$. For every pair of distinct agents $(i, j) \in \mathcal{N} \times \mathcal{N}$, let the meeting action be the pair $\left( a_{ij}, a_{ji} \right) \in [0, 1]^2$. On the other hand, we denote the action of $i$ without the link $a_{ij}$ as

$$\mathbf{a}_{i-(i,j)} = \{a_{ik}, \text{ s.t. } k \neq i, j\} \in [0, 1]^{N-2}.$$

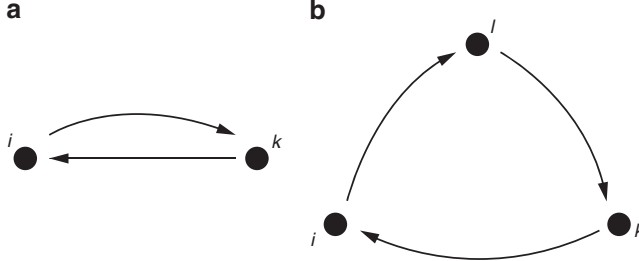

**Fig. 2** Sketch of reciprocity and cycles. **a** The reciprocity of agent $i$ is determined by the sum of the reciprocated ties. **b** The cycles of agent $i$ are computed summing up all the length 3 cycles that starts and ends in node $i$.

Further, let all the actions but the meeting pair $(a_{ij}, a_{ji})$ be

$$\mathbf{a}_{-(i,j)} = \{a_{kl}, \text{ s.t. } k, l \in \mathcal{N}, \ k \neq l, \ (k, l) \notin \{(i, j), (j, i)\}\}.$$

**Definition.** $\mathcal{G}^\star$ *is a pairwise-Nash equilibrium (PNE) if*
**C2.** *for all pairs of agents $(i, j)$,*

$$V_i\left( a_{ij}, \mathbf{a}^\star_{i-(i,j)}, \mathbf{a}^\star_{-i} \right) \leq V_i\left( a^\star_{ij}, \mathbf{a}^\star_{i-(i,j)}, \mathbf{a}^\star_{-i} \right), \ \forall a_{ij} \in [0, 1],$$

$$V_j\left( a_{ji}, \mathbf{a}^\star_{j-(j,i)}, \mathbf{a}^\star_{-j} \right) \leq V_j\left( a^\star_{ji}, \mathbf{a}^\star_{j-(j,i)}, \mathbf{a}^\star_{-j} \right), \ \forall a_{ji} \in [0, 1],$$

**C3.** *for all pairs $(i, j)$, and for all pairs $\left( a_{ij}, a_{ji} \right)$ in $[0, 1]^2$,*

$$V_i\left( a_{ij}, a_{ji}, \mathbf{a}^\star_{-(i,j)} \right) > V_i\left( a^\star_{ij}, a^\star_{ji}, \mathbf{a}^\star_{-(i,j)} \right)$$
$$\Downarrow$$
$$V_j\left( a_{ij}, a_{ji}, \mathbf{a}^\star_{-(i,j)} \right) < V_j\left( a^\star_{ij}, a^\star_{ji}, \mathbf{a}^\star_{-(i,j)} \right).$$

In the above definition, (C2) is the Nash, or selfish, condition which requires that, for every pair $(i, j)$ of agents, neither $i$ nor $j$ can be selfishly better off by changing just her individual outgoing tie. (C2) is similar to (C1). However, (C1) implies (C2) but not vice versa, due to the different action spaces; see Remark 2 and Supplementary Fig. 1 in Supplementary Note 1. On the other hand, (C3) is the cooperative condition, stating that at equilibrium there is no pair of agents who can both be better off by coordinating their actions. Note that (C3) is a Pareto optimality condition (see the Supplementary Note 1).

**Network motifs analysis.** Our results are twofold: in the first analytic part, we initially assume agents being homogeneous ($P_i = P$, $\forall i \in \mathcal{N}$). Thus players' characteristics other than their connections are neglected, and we present a stability analysis in the space of parameters $P = \{\alpha, \beta, \gamma, \delta\}$, of four prototypical network motifs. For each network topology, we analytically derive necessary and sufficient conditions which guarantee NE and PNE. Analytical tools are discussed in the Methods section and in Supplementary Note 1, the formal proofs are available in Supplementary Note 2. In the second part, we consider complex random network models as well as real-world networks, and we assume that actors have heterogeneous (and possibly irrational) behavior: some might be more willing to create close triads to receive social support, whereas others might prefer to build structural bridges in order to get competitive advantages. We then use the Nash stability condition as a quantitative tool to infer the individual preferences $P_i$.

In what follows, we first present the equilibrium analysis of the prototypical network motifs depicted in Fig. 3. The first case is the empty network, i.e., a graph $\mathcal{G}^{EN}$ of $N$ nodes such that for every pair $(i, j)$ of agents, $a_{ij} = a_{ji} = 0$. Given this particular topology, the equilibrium condition must guarantee that no links are initiated.

**Theorem.** *Let $\mathcal{G}^{EN}$ be an empty network. Then*

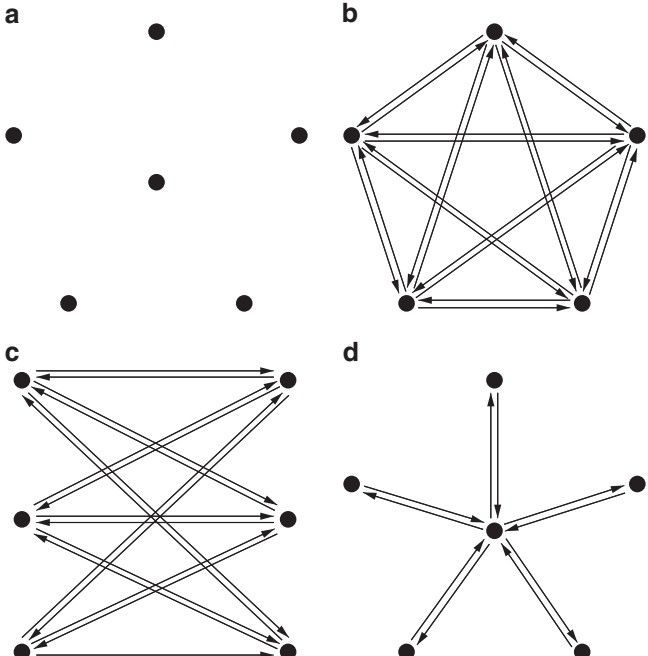

**a**

**b**

**c**

**d**

**Fig. 3** Sketch of the prototypical network motifs studied here. **a** Empty, **b** Complete, **c** Balanced Complete Bipartite, and **d** Star Network. All links shown have unitary weight.

a. $\mathcal{G}^{EN}$ is always a NE,
b. $\mathcal{G}^{EN}$ is a PNE if and only if $\gamma \geq \alpha(1 + \delta + \delta^2)$.

Effectively, when no links are present in the graph, no agent has a selfish incentive to create outgoing ties as she only incurs costs. Thus, the empty network is always a NE. However, when dealing with PNE, cooperation may take place. More precisely, $\gamma \geq \alpha(1 + \delta + \delta^2)$ is the necessary (and sufficient) condition for PNE. For otherwise, the cost parameter is cheap enough that agents would initiate reciprocate ties. In summary, in a stable empty network, either agents are non-cooperative, or creating ties is too costly. Figure 4a shows a comparison of the NE and PNE stability regions of the empty network in the normalized parameter space $\alpha/\gamma$ versus $\beta/\gamma$.

We then move our analysis to the other extremum: the complete network, i.e., a graph $\mathcal{G}^{CN}$ of $N$ nodes such that for every pair $(i, j)$ of distinct agents, $a_{ij} = a_{ji} = 1$. Opposite to the previous case, the equilibrium conditions must guarantee that actors have no incentive on reducing links.

**Theorem**. *Let $\mathcal{G}^{CN}$ be a complete network. Define*

$$\bar{\gamma}_{NE} := \begin{cases} \alpha\delta(1 + \delta(2N - 3)) + \beta(N - 2), & \text{if } \beta > 0 \\ \alpha\delta(1 + \delta(2N - 3)) + 2\beta(N - 2), & \text{if } \beta \leq 0, \end{cases}$$
$$\bar{\gamma}_{PNE} := \alpha\delta(1 + \delta(2N - 3)) + 2\beta(N - 2), \quad \text{then}$$

a. *$\mathcal{G}^{CN}$ is a NE if and only if $\gamma \leq \bar{\gamma}_{NE}$,*
b. *$\mathcal{G}^{CN}$ is a PNE if and only if $\gamma \leq \bar{\gamma}_{PNE}$.*

As one can deduce from the proof (see theorem 8 in Supplementary Note 2), both stability conditions require the cost parameter $\gamma$ to be upper-bounded by the minimal marginal benefit of being linked with everybody else. Concerning the NE, if the ties are too costly ($\gamma > \bar{\gamma}_{NE}$), the best action for each agent is to drop all outgoing ties. This transition behavior has dramatic consequences, as it leads to the empty network if agents were simultaneously playing best response. Similar reasoning applies

for the PNE, albeit agents cannot drop all their outgoing ties simultaneously.

From the comparison between the cost parameter thresholds, we notice that $\bar{\gamma}_{NE} \leq \bar{\gamma}_{PNE}$ with strict inequality if $\beta > 0$. This means that, in some cases, the complete network is pairwise-Nash stable, but it is not Nash stable. Such a discrepancy derives from the different action spaces in the stability notions. However, the comparison in Fig. 4b shows that such a difference is marginally small. Further discussion and a zoomed plot are available in Supplementary Note 2 and Supplementary Fig. 2. Concerning the impact of cooperation in the PNE, the Pareto condition (C3) is always satisfied whenever the selfish condition (C2) is fulfilled. Thus, cooperation does not restrict the PNE stability region for this network motif.

Finally, Fig. 4b evinces that higher values of $\beta$ incentivizes the stability of the complete network. Even when $\alpha = 0$, $\beta \geq \frac{\gamma}{N-2}$ is necessary and sufficient to guarantee NE. A similar condition holds for PNE. We emphasize that this result confirms that complete network stability is correlated with high clustering coefficients, as already shown in refs. [23] and [24]. Yet, it reveals that complete networks are stable even if agents strive for betweenness centrality ($\beta < 0$), provided that they also have a closeness-type incentive ($\alpha/\gamma$ has to be high enough). In other words, it shows that in a stable complete network, ties are relatively cheap and individuals are typically looking for local support, confirming the theory of Coleman[35], as well as trying to improve their social influence.

Next, we consider an interesting class of topologies: the bipartite networks, which are graphs in which the nodes can be partitioned into two factions such that $a_{ij} = 0$ when $i$ and $j$ belong to the same partition. We first analyze the balanced complete bipartite network, where the two factions are of size $N/2$, thus perfectly balanced in the number of nodes, and $a_{ij} = 1$, for all pairs $(i, j)$ of nodes belonging to different partitions (complete). Bipartite networks are interesting case studies, as their equilibrium requires two conditions: (i) existing links across the two partitions must not be dropped, (ii) ties within the same partition must not be created.

**Theorem**. *Let $\mathcal{G}^{BN}$ be a balanced complete bipartite network of $N$ agents, with $N$ being even. Define*

$$\underline{\gamma}_{NE} := \alpha\delta^2\left(\frac{N}{2}\right) + 2\beta\left(\frac{N}{2}\right),$$
$$\underline{\gamma}_{PNE} := \alpha\left(1 + \delta\left(\frac{N}{2} + 1\right) + \delta^2\left(\left(\frac{N}{2}\right)^2 + 3\frac{N}{2} + 1\right)\right) + 2\beta\left(\frac{N}{2}\right),$$
$$\bar{\gamma}_{NE} = \bar{\gamma}_{PNE} := \alpha\delta\left(1 + \delta\frac{N}{2}\right),$$

*then*

a. *$\mathcal{G}^{BN}$ is a NE if and only if $\underline{\gamma}_{NE} \leq \gamma \leq \bar{\gamma}_{NE}$,*
b. *$\mathcal{G}^{BN}$ is a PNE if and only if $\underline{\gamma}_{PNE} \leq \gamma \leq \bar{\gamma}_{PNE}$.*

According to the theorem, $\gamma \leq \bar{\gamma}_{NE}$ guarantees that agents have no incentive on dropping existing links and $\gamma \geq \underline{\gamma}_{NE}$ ensures that agents creating links within the same partition will incur sufficiently high cost. Analogous conditions hold for PNE. Figure 4c shows that, regardless of the stability notion, the second effect emerges to be more relevant as it excludes the upper part of the diagram from the stability region. This confirms that balanced complete bipartite network stability is negatively correlated with the clustering coefficient $\beta$, hence positively correlated with betweenness centrality, as previously found in ref. [24]. Thus, we can infer that in a stable balanced complete bipartite network actors are competitive in looking for brokerage

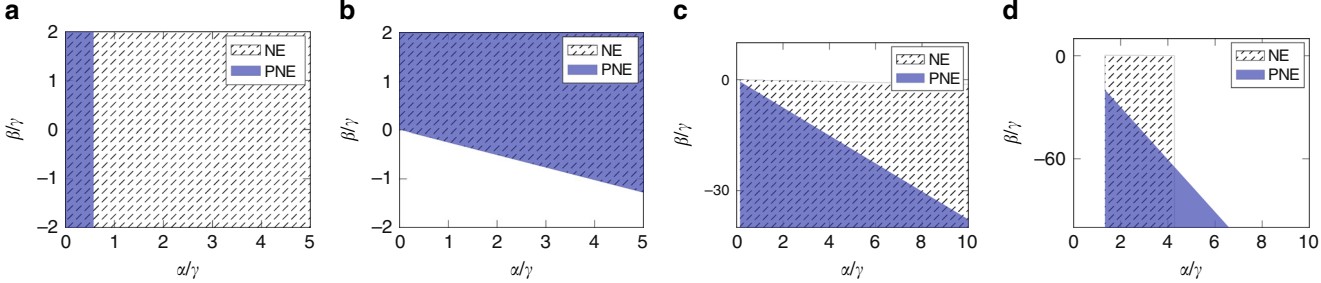

**Fig. 4** NE and PNE regions of the four network motifs of homogeneous agents. **a** Empty, **b** Complete, **c** Balanced Complete Bipartite, **d** Star Network. All the plots refer to $N = 50$ agents and $\delta = 0.5$. Different settings lead to qualitatively similar results.

opportunities and links are relatively cheap as the network is dense. A comparison between NE and PNE stability regions indicates that the balanced complete bipartite network is less likely to be stable when agents can cooperate. As the proof evinces (see Theorem 9 in Supplementary Note 2), actors belonging to the same partition may find it beneficial to establish a mutual tie.

We finally analyze the star network, which can be viewed as an unbalanced complete bipartite network.

**Theorem.** *Let $\mathcal{G}^{SN}$ be a star network. Define*

$$\underline{\gamma}_{NE} := \max\left\{\alpha\delta^2 + 2\beta, \alpha\delta\left(\delta - \frac{1}{N-3}\right)\right\},$$

$$\underline{\gamma}_{PNE} := \alpha\left(1 + 2\delta + (N+3)\delta^2\right) + 2\beta,$$

$$\bar{\gamma}_{NE} = \bar{\gamma}_{PNE} := \alpha\delta(1 + \delta),$$

*then*

a. $\mathcal{G}^{SN}$ *is a NE if and only if* $\underline{\gamma}_{NE} \leq \gamma \leq \bar{\gamma}_{NE}$,
b. $\mathcal{G}^{SN}$ *is a PNE if and only if* $\underline{\gamma}_{PNE} \leq \gamma \leq \bar{\gamma}_{PNE}$.

The proof evinces that equilibrium requires three conditions: (i) the central node must have no incentive in dropping her ties, (ii) the periphery nodes must not destroy the link to the center of the star, and (iii) must not initiate ties among them. However, whenever (ii) is satisfied, (i) follows, thus it reduces to two conditions. As shown in Fig. 4d, stability requires a lower bound on $\alpha/\gamma$ and a linear upper bound on $\beta/\gamma$ with respect to $\alpha/\gamma$, which, respectively, guarantee (ii) and (iii). Moreover, in the NE stability region, an additional upper bound on $\alpha/\gamma$ prevents from being beneficial for a peripheral agent to drop the link to the central one, while simultaneously connecting to all her similars. Such a threshold is not present in the PNE stability region, as deviations are allowed only by pairs of agents. Further discussion is addressed in Supplementary Note 2 and in Supplementary Figs. 3 and 4.

Ultimately, we discovered that stability of the star network is correlated with high values of the influence parameter $\alpha$, and thus with closeness centrality incentives as in ref. [22]. Yet, our proof suggests that the situation is more complex: we found that observing a stable star network indicates that actors are not interested in local social support and that ties are costly as the star is sparser than the balanced complete bipartite.

**Phase diagram comparison.** So far we have seen the correlation between different incentives and the stability regions of different network architectures. Now we focus on the co-existence of equilibria in the parameter space of actor preferences. Figure 5a shows that empty, complete, star and balanced bipartite networks are found to overlap in certain regions of the parameter space. Conversely, Fig. 5b shows that bipartite and complete networks cannot be simultaneously pairwise-Nash stable, due to

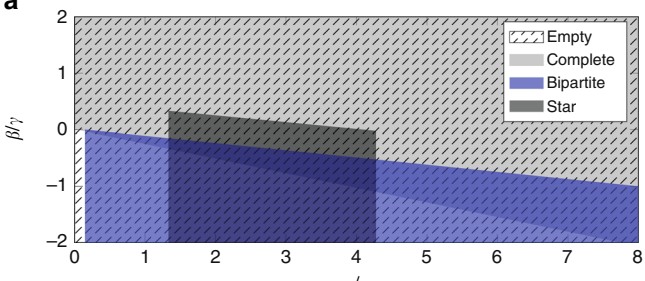

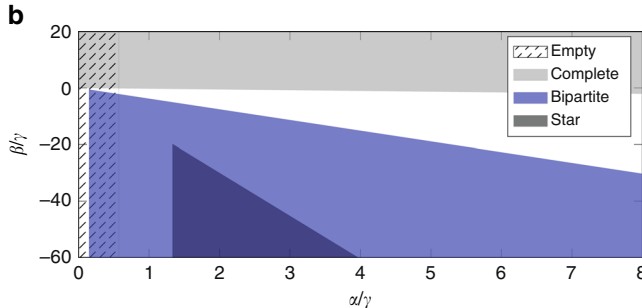

**Fig. 5** Phase Diagram comparison. **a** Nash and **b** Pairwise-Nash equilibrium comparisons of different network motifs of $N = 50$ agents and for $\delta = 0.5$.

cooperation. Similarly, the overlap between the pairwise-Nash stability region of complete and empty networks is now restricted mainly in the proximity of the origin of the phase diagram, i.e., where the cost parameter $\gamma$ is high. Yet, both plots show that the clustering coefficient $\beta$ draws a significant separation between complete and bipartite networks.

**Inference of behavior for complex networks.** The theoretical analysis developed so far is elegant and powerful, though restricted to stylized models that appear merely as motifs in empirical networks. In order to be able to analyze complex networks, we need to introduce two elements: (i) agents' heterogeneity, and (ii) irrationality in the form of an error term. Our goal in this section is to provide the most rational estimate of the heterogeneous individual parameters from an observed state of the network, e.g., a Nash equilibrium. This learning and inference problem can be cast as an inverse optimization problem over candidate objective functions. In econometric and operations research, the common approach is called structural estimation, a method that relies on the existence of a set of necessary (structural) equations for unknown parameters, e.g., first-order optimality conditions for convex problems[46]. Due to the complexity of our structural equations, we developed our own

behavior estimation method, borrowing some ideas from ref. [47]. The complete description can be found in the Methods and in Supplementary Note 3. Below, we summarize the main steps.

Firstly, we use the alternative formulation of the payoff function, introducing a new set of individual parameters

$$\boldsymbol{\theta}_i = \begin{bmatrix} \theta_{i,1} \\ \theta_{i,2} \\ \theta_{i,3} \end{bmatrix} = \begin{bmatrix} \alpha_i \delta_i / \gamma_i \\ \alpha_i \delta_i^2 / \gamma_i \\ \beta_i / \gamma_i \end{bmatrix} \in \Theta = \mathbb{R}_{\geq 0} \times \mathbb{R}_{\geq 0} \times \mathbb{R}.$$

Our choice is motivated by the socio-theoretical interpretation of the influence measure $t_i$. We emphasize that the normalization by $\gamma_i$ is without loss of generality as the equilibria do not change. Moreover, the original set of parameters $P_i$ can be recovered, though $\delta_i$ is not necessarily in $[0, 1]$.

Drawing inspiration from the Nikaidô–Hisoda function[48], we introduce the error function

$$e_i(\mathbf{a}_i, \boldsymbol{\theta}_i) := V_i(\mathbf{a}_i, \mathbf{a}^\star_{-i}, P_i) - V_i(\mathbf{a}^\star_i, \mathbf{a}^\star_{-i}, P_i),$$

which takes positive values whenever the preference $\boldsymbol{\theta}_i$ is such that the Nash equilibrium constraint is violated by the action $\mathbf{a}_i$ of agent $i$. Hence, if we consider only the positive contributions $e_i^+(\mathbf{a}_i, \boldsymbol{\theta}_i) := \max\{0, e_i(\mathbf{a}_i, \boldsymbol{\theta}_i)\}$, we can define the average Euclidean distance from the NE conditions as

$$d_i(\boldsymbol{\theta}_i) := \left( \int_{\mathcal{A}} e_i^+(\mathbf{a}_i, \boldsymbol{\theta}_i)^2 \, d\mathbf{a}_i \right)^{1/2}.$$

In our behavior estimation method the goal is to determine the individual set of preferences $\hat{\boldsymbol{\theta}}_i$ which minimize the distance function $d_i(\boldsymbol{\theta}_i)$. The Minimum NE-Distance problem, which might not have a unique solution, can be solved exactly when

there exists a non-empty region $\Theta_{i,0}$ in the parameter space where no NE violations occur:

$$\Theta_{i,0} := \{ \boldsymbol{\theta}_i \in \Theta, \text{s.t.} \, \forall \mathbf{a}_i \in \mathcal{A}, e_i^+(\mathbf{a}_i, \boldsymbol{\theta}_i) = 0 \}.$$

Conversely in case of NE violations (e.g., due to bounded rationality or noisy observations), it can be solved as an Ordinary Least Square (OLS) problem. This allows us to identify an OLS estimate, to unbias it, and to compute confidence intervals.

In the following, we show the applicability of our behavior estimation method to two well-known real-world datasets, as well as two celebrated random networks models.

**Medici network.** The first example concerns the network of marriage and business connections among Florentine families in the fifteenth century, originally collected by Kent[49], but first coded by Padgett and Ansell[41]. Unlike more common undirected renditions (see[50]), we represent this network as a directed multiplex graph consisting of marriage and business ties (see Fig. 6), based on the works of refs. [51] and [52], respectively. A red arc from $i$ to $j$ represents a female from family $i$ married into family $j$, whereas blue ties point towards the most prosperous family. We applied our estimation method to three different scenarios: (i) marriage ties network, (ii) business ties network, and (iii) combined network.

We focus our analysis on the Medici, who were able to rise in power, even though Florence was previously ruled by an oligarchy of elite families. According to Padgett and Ansell[41], a key to understanding this lies in the structure of social network relationships. The Medici gradually but surely exploited the structural holes within the oligarchic marriage network, and enabled structural isolation among Medici partisans, deterring

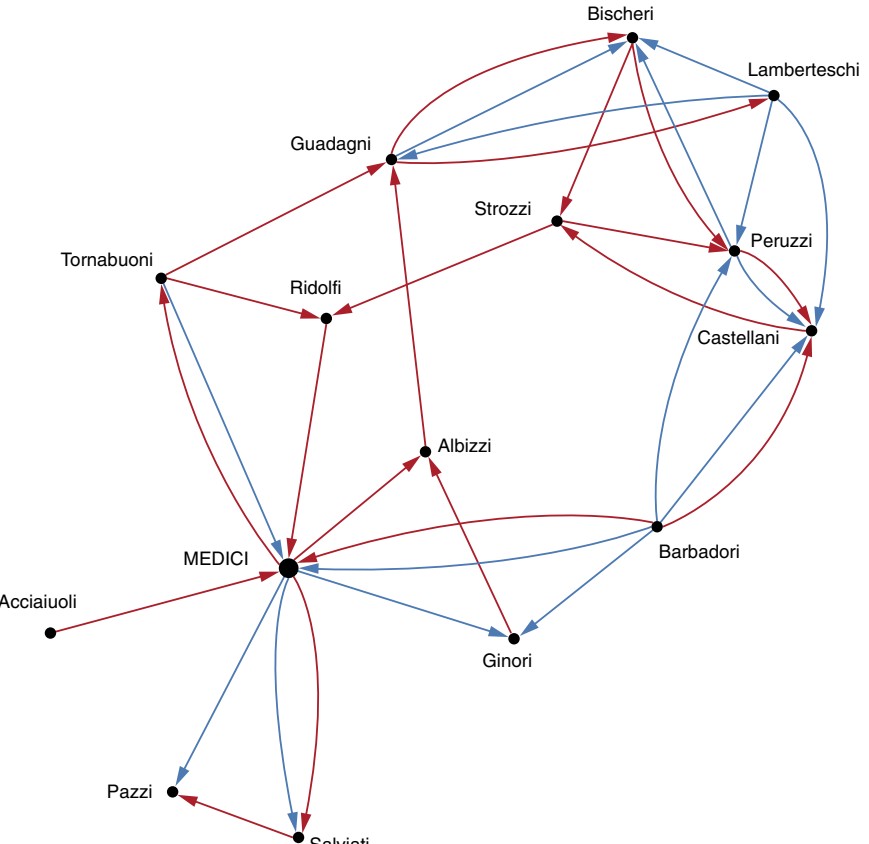

**Fig. 6** Multiplex graph of marriages (red) and business (blue) relations between 15 Florentine families. The adjacency matrix of the combined network is constructed as the maximum between the adjacency matrices that correspond to the marriage and the business networks.

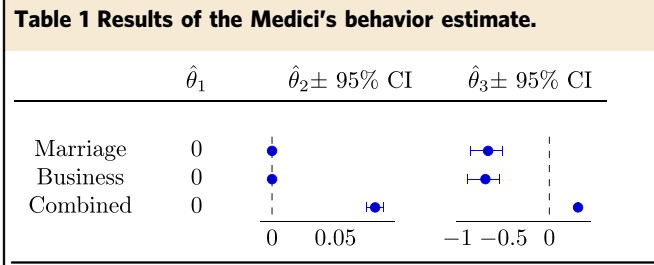

**Table 1 Results of the Medici's behavior estimate.**

We consider three different network settings, (i) Marriage, (ii) Business, and (iii) Combined. The estimate of the reciprocity parameter $\hat{\theta}_1$ hits the non-negativity constraint, thus it is shown without Confidence Interval (CI). The same comment applies to the estimate of $\hat{\theta}_2$ in the Marriage and Business settings. The Confidence Intervals are built according to the method described in Supplementary Note 3, using a regular mesh of $n = 2^{N-1}$ samples in the action space $A$, where $N = 15$ agents. Source data are provided as a Source Data file

them from marrying or having business with the oligarchs. Yet, Padgett and Ansell pointed out that the Medici party was a very centralized system consisted almost entirely of direct ties to the Medici family, and that, within their own party, the Medici did not marry those families with whom they engaged in economic relations, nor did they do business with those whom they married.

In order to understand the structural isolation operated by the Medici family, we compare the outcome of our behavior estimation method in the three different settings (see Table 1): the negative values of $\hat{\theta}_3$ in the isolated settings (marriage or business) testify the broker position of the Medici family in both contexts, also in agreement with the analysis of the brokerage coefficients in the isolated marriage and business networks by Sims and Gilles[51] and Ostrom and Crothers[52]. On the other hand, though, the analysis of the combined network evinces the emergence of the opposite behavior with respect to the parameter $\hat{\theta}_3$. Thus, the structural isolation operated on multiplex ties not only guaranteed stability (preventing dissent spreading) but at the same time enhanced social (and political) support (positive $\hat{\theta}_3$) to the Medici family. Furthermore, note that reciprocity cannot emerge in the isolated settings because of the way relations are described in the data. However, the estimate of $\hat{\theta}_1$ in the combined network confirms the segregation of types of ties with the Medici itself observed by Padgett and Ansell. A low estimate of $\hat{\theta}_2$, finally, can be associated with the scarce tendency to cycles which would reduce the segregation of the leaves in the centralized network system of the Medici party.

**Australian bank dataset.** The second example deals with a study of structure in a number of branches of a large Australian bank[42]. The unweighted directed network of relationships shown in Fig. 7 is from one particular branch in response to the question "In whom do you feel you would be able to confide if a problem arose that you did not want everyone to know about?", i.e., the con- fiding relations, though the same study also analyzed the advice- seeking, close friendship and satisfying interactions relations. From the attributes of the nodes, four different hierarchical levels can be distinguished, i.e., Branch Manager, Deputy Manager, Service Adviser and Teller. Low hierarchical positions occupy the periphery of the network, while high ranked nodes have more incoming connections. From a macroscopic inspection, one can detect the presence of star-like motifs embedded in the network, e.g., around the Branch Manager, and the Service Advisers 1 and 2. However, no other motifs are discernible.

The outcome of our behavior estimation method is shown in Fig. 8. From the analysis one evinces that more competitive behaviors (negative values of $\hat{\theta}_3$) are typical of high hierarchical

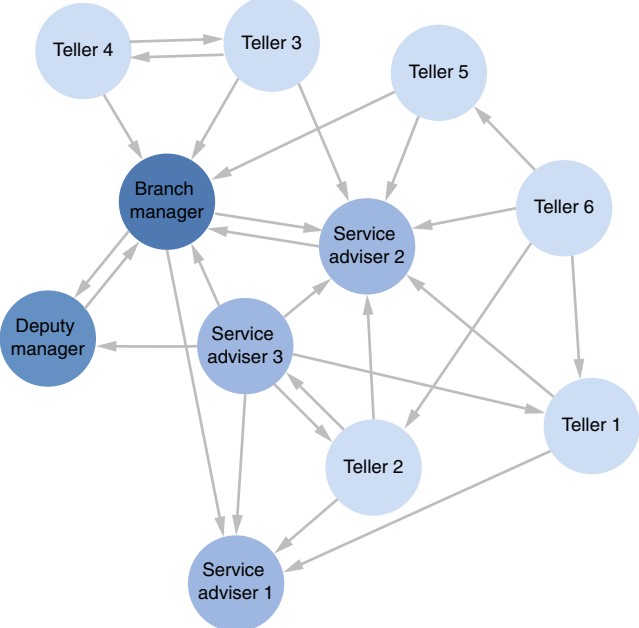

**Fig. 7** The network of confiding relationship among the 11 agents of the Australian bank dataset[42]. Every link has unitary weight and more important nodes have darker color.

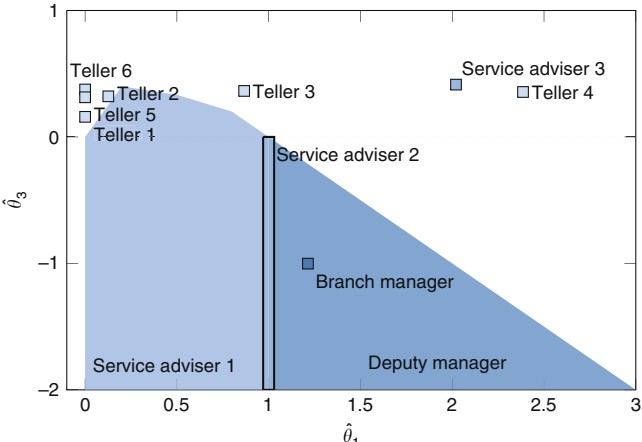

**Fig. 8** Best estimates $(\hat{\theta}_1, \hat{\theta}_3)$ for the Australian bank dataset[42]. A detailed description of the results and the confidence interval analysis are available in Supplementary Note 4 and Supplementary Table 2.

positions, e.g., Branch and Deputy manager. Conversely, low- ranking positions are more inclined towards social support (positive $\hat{\theta}_3$), as witnessed by the behavior of tellers 1–6. As observed by Pattison[42], confiding relations are likely to be more local or restricted in their span, linking individuals from one level in the organization to those in the next. Thus, it is unlikely that high-rank agents exhibit clustering behavior, as there are fewer nodes in the top level of the hierarchical structure.

The complete analysis reported in Supplementary Note 4 also shows that agents are not particularly inclined towards cyclic structures, in accordance with Davis[36] who showed that cycles are atypical structures in hierarchical networks. Finally, notice that several actors at different hierarchical levels, i.e., Branch and Deputy Managers, Service Adviser 2 and 3, Tellers 3 and 4 exhibit

relatively high values of $\hat{\theta}_1$, which is symptomatic of higher reciprocity in the relationships. We emphasize that, since relations that span across different hierarchical levels are not common, it is likely for high-rank agents to reciprocate ties, as they have fewer options.

**Complex random networks**. As discussed in the introduction, a large class of network formation models is based on probabilistic mechanisms. The aim of this section is to use our behavior estimation method to give a sociological and strategic interpretation of them.

The first example concerns the celebrated preferential attachment model introduced by Barabási and Albert[7]. In our analysis, we draw inspiration from ref. [53] to construct a directed version of the model where newborn nodes, introduced over time, receive $m_{in} = 2$ incoming edges from existing nodes (selected proportionally to their outdegree) and build $m_{out} = 2$ outgoing ties, whose receivers are now selected proportionally to their indegree. The resulting networks exhibit scale-free properties in both in- and outdegree measures (see Supplementary Fig. 6).

As we are looking at a growing process, we study the outgoing ties of the newborn agents $n$ as soon as they are introduced to the network of $n - 1$ agents. More specifically, we consider 50 realizations of the Preferential Attachment growing process, and for each of them we focus our analysis on eight newborn agents $n$, as shown in Fig. 9. When the network size $n$ is still small, newborns tend to reciprocate links and form clusters (red values are symptomatic of high values of $\hat{\theta}_1$ and $\hat{\theta}_3$). When the network size grows, the behavior of the newborns becomes more and more adverse to reciprocity and clustering. Yet, the tendency of forming reciprocal ties and clusters does not entirely vanish.

In our second example, we consider the small-world model introduced by Watts and Strogatz[5], and we adapt it by rewiring only the outgoing edges, as in[54]. In Fig. 10 we collect the heat maps of the average distance function for several networks of fixed size $N = 11$, different outdegree $k$, and rewiring probability $p$. We emphasize that when $p = 0$ the network is a regular ring lattice where each node is mutually connected with the $k$ closest nodes in both directions. In these cases (first row of the figure), the increase of $k$ coincides with an increase of the local clustering coefficient, which, in turns, corresponds to a change of the estimate of the clustering coefficient $\hat{\theta}_3$ from the negative to the positive half-space. On the other hand, when $k = (N - 1)/2$, the ring lattice is a complete network and the rewiring process has no influence. As a matter of fact, the plots of the last column are all equal and they confirm the theoretical results of the complete network (see Fig. 4b). In the other cases, increasing the rewiring probability $p$ shifts the estimates of the reciprocity $\hat{\theta}_1$ towards increasingly smaller values. Such a phenomenon depends on the fact that the rewiring process tends to destroy the initial symmetry of the ties in the ring lattice.

To conclude, the behavior estimation method reveals the existence of a pattern behind random network models, allowing for sociological and strategic interpretations of the probabilistic mechanisms. Further details are available in Supplementary Note 4.

## Discussion

We proposed a parametric model of strategic network formation where actors can control their followees but not their followers, and they are endowed with a novel payoff function, which is a parametric combination of different incentives: influence, brokerage, closure. In the theoretical analysis, agents are assumed homogeneous, rational, and myopic. We analytically derived

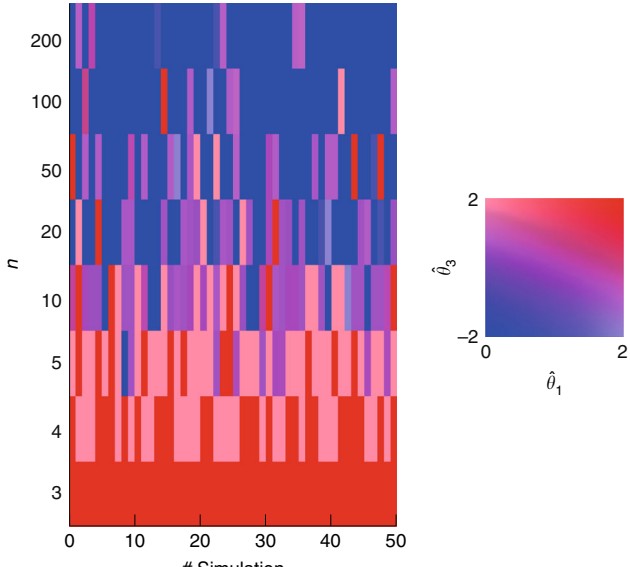

**Fig. 9** Results of the Preferential Attachment test. For each of the 50 simulations, we study the behavior of the newborn agent $n$. The left plot shows the minimizer of the distance function through the color map on the right, which associates a color to each pair $(\hat{\theta}_1, \hat{\theta}_3)$. For illustration and computational purposes, we restrict the parameter space to $\tilde{\Theta} = [0, 2] \times 0 \times [-2, 2]$, and the action space of the newborn agent $n$ to $\tilde{\mathcal{A}} = \{\mathbf{a}_n \in \{0, 1\}^{n-1}, \text{ s.t. } ||\mathbf{a}_n||_1 \leq m_{out}\}$. In other words, the newborn agent $n$ can only choose among all the combinations of (at maximum) $m_{out}$ outgoing edges (of weight 1). Hence, we solve $(\hat{\theta}_{n,1}, \hat{\theta}_{n,3}) \in \arg\min_{\theta_n \in \tilde{\Theta}} \left( \int_{\tilde{\mathcal{A}}} e_n^+ (\mathbf{a}_n, \theta_n)^2 \, d\mathbf{a}_n \right)^{1/2}$, and we select the estimate with maximum Euclidean norm among the minimizers.

stability conditions of several network motifs, considering purely selfish and selfish-cooperative scenarios. We confirmed existing results on the correlation between incentives and stable network architectures, yet revealing new transition paradigms and opening the door for future investigation on cascade effects and robustness of equilibria. Further, our closed-form analytic stability conditions depend, in a parametric fashion, on the trade-offs between different individual incentives, thus providing a way to infer individual tendencies from the observed stable networks. Yet, we quantitatively describe the impact of cooperation on network stability.

We also considered complex networks scenarios with heterogeneous and not necessarily rational actors. Using the Nash equilibrium condition we constructed a statistical behavior estimation method capable to learn the individual user preferences. We applied this method to real-world datasets and random networks models, providing evidence that our results cross-validate empirical, historical, and sociological observations and our method offers sociological and strategic interpretation of random networks mechanisms. We emphasize that our model can be adapted to different descriptions of the payoff function, e.g., considering an extra cost for changing ties, or other individual incentives such as eigenvector centrality, or constraining competitors' brokerage, as well as to different definitions of equilibrium, e.g., mixed-Nash equilibria or continuous parametric transitions of selfish-cooperative behavior as in[55].

## Methods

**Theoretical analysis of network motifs**. The goal of our theoretical analysis is to derive necessary and sufficient parametric conditions which guarantee NE/PNE. Concerning the NE conditions, we turn the definition into an optimization problem: $\mathcal{G}^*$ is a Nash equilibrium if, for all agents $i$, $\mathbf{a}_i^* \in \arg\max_{\mathbf{a}_i \in \mathcal{A}} V_i(\mathbf{a}_i, \mathbf{a}_{-i}^*)$.

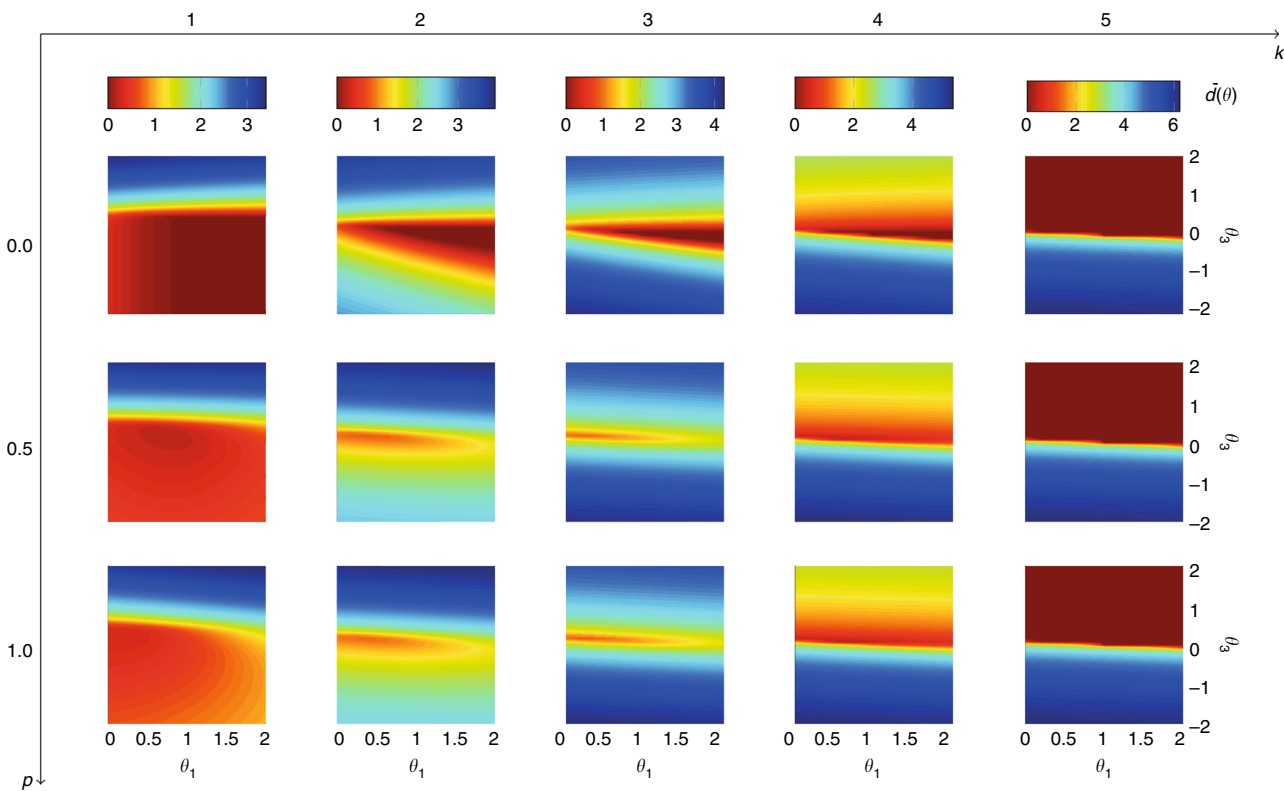

**Fig. 10** Results of the test on Small-World networks of fixed size $N = 11$ agents, different values of outdegree $k$, and of rewiring probability $p$. For each pair $(k, p)$ we consider $n = 10$ different realizations and we compute the average distance function $\tilde{d}(\boldsymbol{\theta}) = \frac{1}{nN} \sum_{i=1}^{n} \sum_{j=1}^{N} d_i^j(\boldsymbol{\theta})$, where $d_i^j(\boldsymbol{\theta})$ is the distance function of agent $i$ in the $j$th realization. For illustration and computational purposes, we restrict the parameter space to $\tilde{\Theta} = [0, 2] \times 0 \times [-2, 2]$. The plot shows the heat map of $\bar{d}(\boldsymbol{\theta}) = \log(\tilde{d}(\boldsymbol{\theta}) + 1)$ in the $(\theta_1, \theta_3)$ space, with lower values (thus more likely) associated with the red color. Source data are provided as a Source Data file.

Then, we use the Variational Inequality (VI) approach (discussed in Supplementary Note 1) in order to derive necessary conditions.

**Theorem.** *If $\mathcal{G}^\star$ is a Nash equilibrium then*

$$\langle \nabla_{\mathbf{a}_i} V_i\left(\mathbf{a}_i, \mathbf{a}^\star_{-i}\right)\big|_{\mathbf{a}^\star_i}, \ \mathbf{a}_i - \mathbf{a}^\star_i \rangle \leq 0, \quad \forall \mathbf{a}_i \in \mathcal{A}, \quad \forall i \in \mathcal{N}.$$

These conditions are then checked for sufficiency. If not, we restrict them by means of educated counterexamples, and verify again.

Concerning the PNE, we use an analogous VI approach in order to derive necessary conditions for the Nash property (C2). In this case, the necessary conditions are shown to be also sufficient. Finally, concerning the Pareto optimality property (C3), we obtain the necessary conditions by imposing (C3) for appropriate meeting actions. To guarantee sufficiency, we invoke the following result.

**Theorem.** *Let $\mathcal{G}^\star$ be a network.*

(i) *If (C2) is satisfied for a pair of agents $(i, j)$ and $a^\star_{ij} = a^\star_{ji} = 1$, then (C3) is satisfied for the same pair;*

(ii) *if for all pairs $(i, j)$, the pair $(a^\star_{ij}, a^\star_{ji})$ satisfies for all pairs $(a_{ij}, a_{ji}) \in [0, 1]^2$,*

$$V_i\left(a_{ij}, a_{ji}, \mathbf{a}^\star_{-(i,j)}\right) + V_j\left(a_{ij}, a_{ji}, \mathbf{a}^\star_{-(i,j)}\right) \leq V_i\left(a^\star_{ij}, a^\star_{ji}, \mathbf{a}^\star_{-(i,j)}\right) + V_j\left(a^\star_{ij}, a^\star_{ji}, \mathbf{a}^\star_{-(i,j)}\right),$$

*then the Pareto Optimality condition (C3) is satisfied.*

**Behavior estimation method.** For the behavior estimation method, we introduce the NE-distance function $d_i(\boldsymbol{\theta}_i)$, and the goal is to identify the set of parameters $\hat{\boldsymbol{\theta}}_i$ that minimizes it. In order to approach the problem, we first verify that it is well-posed by proving convexity and smoothness of the distance function (the proof can be found in Supplementary Note 3). These two properties imply both tractability as well as scalability of algorithmic approaches to the Minimum NE-Distance problem.

**Theorem.** *Let $f(\mathbf{x}, \boldsymbol{\theta}) : \mathbb{R}^n \times \mathbb{R}^p \to \mathbb{R}$ be a continuous function of $\mathbf{x} \in \mathbb{R}^n$ and $\boldsymbol{\theta} \in \mathbb{R}^p$. Moreover, assume $f$ to be linear in $\boldsymbol{\theta}$, and let $\mathcal{X}$ be a compact subset of $\mathbb{R}^n$.*

*Consider the following function:*

$$F(\boldsymbol{\theta}) := \int_{\mathcal{X}} (\max\{0, f(\mathbf{x}, \boldsymbol{\theta})\})^2 \mathrm{d}\mathbf{x}.$$

*Then F is continuously differentiable, and its gradient is*

$$\nabla_{\boldsymbol{\theta}} F(\boldsymbol{\theta}) = \int_{\mathcal{X}} 2 \nabla_{\boldsymbol{\theta}} (f(\mathbf{x}, \boldsymbol{\theta})) \max\{0, f(\mathbf{x}, \boldsymbol{\theta})\} \mathrm{d}\mathbf{x}.$$

*Moreover, F is a convex function.*

Then, we distinguish two cases. In the first case there are no violations, and the minimum of the distance function is 0. In this case, the set of minimizers is a convex polyhedron that can be described by a finite number of inequalities:

$$\Theta_{i,0} := \left\{ \boldsymbol{\theta}_i \in \Theta, \text{s.t.} \, \forall \mathbf{a}_i \in \mathcal{A}_{\{0,1\}}, e_i(\mathbf{a}_i, \boldsymbol{\theta}_i) \leq 0 \right\},$$

where $e_i(\mathbf{a}_i, \boldsymbol{\theta}_i) = V_i\left(\mathbf{a}_i, \mathbf{a}^\star_{-i}, P_i\right) - V_i\left(\mathbf{a}^\star_i, \mathbf{a}^\star_{-i}, P_i\right)$ and $\mathcal{A}_{\{0,1\}} = \{0, 1\}^{N-1}$, i.e., the inequalities need to be evaluated only at the vertices of the action space. We emphasize that, for all $\boldsymbol{\theta}_i \in \Theta_{i,0}$, the NE conditions corresponding to agent $i$ are full-filled.

In the second case, when NE violations occur, we consider a discrete version of the Minimum NE-Distance problem by approximating the integral over the action space with a finite sum over a regular grid. The discrete problem inherits smoothness and convexity, so it is possible to use a projected gradient method to find a solution. Moreover, the discrete problem can be viewed as an Ordinary Least Square regression problem, with the exception that error terms are non-negative, thus the bias of the estimates is non-zero and errors cannot be modeled as normally distributed. Nonetheless, we are able to build confidence intervals of the parameters. The detailed method is described in Supplementary Note 3.

**Reporting summary.** Further information on research design is available in the Nature Research Reporting Summary linked to this article.

## Data availability

The datasets as well as the source data files discussed are available, respectively, at the following public repositories[56,57]. A reporting summary for this article is available as a Supplementary Information file.

## Code availability

The code that performs the behavior estimation method as well as the tests of the datasets and of the random network models are available at the following public repository[58].

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

## Acknowledgements

The authors thank Prof. C. Stadtfeld for the interesting discussions on the individual incentives, and M. Gallana for his contribution on the statistical interpretation of the behavior estimation method. They also gratefully acknowledge financial support from ETH Zürich.

## Author contributions

N.P. and F.D. designed research, N.P. performed research, N.P. wrote this manuscript, and F.D. edited this manuscript.

## Competing interests

The authors declare no conflict of interest.
