## [Peer Review File · Nature Communications]

Reviewers' Comments:

Reviewer #1:

Remarks to the Author:

The paper "Social network formation: from systemic stability to individual incentives" studies the creation of social networks making use of a model based on game theory. In particular, the pay-off of the different agents is a trade-off between different topological quantities that describe influence, efficacy and safety, and two different stability definitions are used (Nash equilibrium and pairwise-Nash equilibrium). The model confirms some results already seen in the literature and yields some original conclusions, and it is used to describe a real case. The methodology studies weighted and directed networks, which improves its applicability.

The paper is well written, it is clearly explained, the authors have developed extensive analytical and numerical work, and the subject under study could be of high interest for researchers working on social and economic networks. The figures are correctly done.

While I believe in the potential general interest of this paper, my main complaint is that in the present manuscript the wide impact of the results is not sufficiently described. For this reason, some of my main comments are directed to enlarge the potential influence and impact of the work in the community. In particular, sociology, economics and network theory results have very often been very disconnected from each other, with no interaction among the three different communities. In fact, too many times the same results have been found and developed in parallel in these three different contexts. As Nature Communications is devoted to a wide variety of potential readers, this is a very good opportunity to collaborate to solve this endemic problem of the field.

Attending to this search for generalization, there are some main questions and some other minor details that should be clarified or improved before I can recommend the publication of this paper in Nature Communications.

My first complaint is that the paper, especially in the introduction, has focused on the work developed on the sociology and economics literature (mainly in the latter), but has largely ignored most work done in the context of complex networks, which is more general and is also extensive in the field (Castellano et al., 2009). I address the authors to start including (at least in the introduction) a detailed state of the art of social network formation from the general perspective of network science/complexity. In fact, please note that the literature cited in the socio-economics context in the paper is relatively old (mainly before 2012), which could show that the subject is losing importance in the last decade. A thorough analysis of more general literature will show that this is not the case, and that the field is quite "alive" in the network science context.

I agree with the authors in the fact that there is still no common agreement on the type of centrality that should be used when studying the evolution/creation of networks. However, in the last decade some quantities not cited in the paper have resulted to be very useful in this context (another consequence perhaps of the disconnection between disciplines mentioned above). Just an example: the eigenvector centrality has been widely used to measure knowledge associated to models of innovation and culture spreading (König et al, 2008), the probability of being infected in epidemiology (Newman, 2010), or the importance of webpages (Langville, 2006). It has also been recently applied in the game theory context in this journal (Iranzo et al., 2016), where several networks compete for centrality associated to Nash equilibria. Furthermore, there is already experimental proof of the prevalence of the eigenvector centrality over other topological measures in real systems: In (Banerjee, 2013) microfinance participation in rural Indian villages was shown to be significantly higher when the individuals used as injection points showed higher eigenvector centrality. In

summary, please compare the results obtained by using the novel definition of the pay-off presented in this paper with a more general description of the field.

One of the main targets of the paper is to show that combining more than one topological quantity in the pay-off of the nodes when describing strategic network formation can reflect the real phenomenology more accurately than with a single one. In my opinion this is a suggestive idea that deserves the thorough analysis that this paper develops. However, and as the authors affirm, a similar question was already described in [15], where both betweenness and closeness were combined. Also, in (Grauwin et al, 2009) a pay-off was introduced as a continuous interpolation between cooperative and individual dynamics. Then, which are the main advantages of this new methodology in comparison to the already existing ones?

Two types of equilibria are studied, Nash equilibrium and pairwise-Nash equilibrium. This is an interesting way to measure purely selfish and partially coordinated behaviors in the agents. However, I would appreciate some more information related to the actions that each node is allowed to do in order to characterize more precisely the Nash equilibria. In particular, can each node change the weight of any of its outgoing links an arbitrarily large or small quantity, create links with any weight, etc? If this is the case, then the system would face mixed Nash equilibria, which enables a probabilistic analysis of the system and increases its applicability. Please describe this question with some more detail.

In the manuscript, four very basic types of networks were analytically studied. This is interesting because theoretical work is a powerful tool, but while most work on economic models are restricted to cliques and stars of different sizes, it is known that most real R&D networks are sparse, locally dense and show heterogeneous degree distributions (Cowan, 2004; Powell et al., 2005). In general, networks with different topology behave very differently. I would appreciate a generalization (perhaps only numerical) to more realistic networks (scale-free networks, random ER, small-world, or regular networks, to cite just a few) to show the real applicability of the methodology. This would increase the impact of the paper substantially, and it would become attractive for scientists beyond the economic community.

In several figures there are parameter regions where different Nash equilibria co-exist. Would it be possible to study the transitions from one equilibrium to the rest? That is, if one node decided to change its connection (obviously losing some pay-off), would the rest be obliged to change also their connections and push the system towards a different equilibrium? This analysis would yield valuable information about the system, such as whether some equilibria are more stable than the rest (Iranzo et al, 2016).

In summary, if the authors address my comments and complaints with the aim of increasing the impact and applicability of the results presented in their manuscript I will be happy to recommend the paper for publication in Nature Communications.

Minor comments:

In the abstract the out-of-equilibrium dynamics of the system is cited, but as far as I have read the work focuses on Nash equilibria. Please clarify this question.

In Fig. 5 the text affirms that there are coexistence of all four types, but I can't see it in the figure.

In Fig. 7 the Australian bank data set was analyzed as a Nash equilibrium. Why should the system have already reached such equilibrium? Please explain in more detail.

When describing Nash equilibrium (NE), the authors cite LinkedIn connections as an example. However, in LinkedIn every two nodes connected by a link must agree to create such link while in the NE of the paper each agent totally controls its own outgoing links. Please clarify.

Please define Pareto optimality condition and Pareto optimal front for completeness.

References:

- C. Castellano, S. Fortunato and V. Loreto, *Rev. Mod. Phys.* 81, 591 (2009).
M.D. König and S. Battiston, *From Graph Theory to Models of Economic Networks. A Tutorial* (Springer Berlin Heidelberg, Berlin, Heidelberg, 2009), pp. 23–63.
M. E. J. Newman, *Networks: an introduction* (Oxford Univ. Press, 2010).
C. Langville and A.N. Meyer, *Google's PageRank and Beyond: The Science of Search Engine Rankings* (Princeton University Press, 2006).
J. Iranzo, J. M. Buldú and J. Aguirre, *Nature Communications* 7, 13273 (2016).
A. Banerjee, A. G. Chandrasekhar, E. Duflo and M. O. Jackson, *Science* 341 (2013).
S. Grauwil, E. Bertin, R. Lemoy and P. Jensen, *Proc. Natl Acad. Sci. USA* 106, 20622–20626 (2009).
R. Cowan, *Research Memoranda 016*, Maastricht : MERIT, Maastricht Economic Research Institute on Innovation and Technology. (2004)
W.W. Powell, D.R. White, K.W. Koput and J. Owen-Smith, *American Journal of Sociology* 110:1132–1205 (2005).

Reviewer #2:

Remarks to the Author:

I really like the framework described in this paper, which enables studying and comparing a number of possible incentives that actors can have when forming their networks. Besides its theoretical elegance, a major benefit of the framework is that it has the potential to be used to infer actual individual incentives from real-world stable network architectures. This can be very valuable, as many elegant models can be devised but not many are actually useful.

However, the empirical part of the paper is currently somewhat glossed over. The data set used is scarcely described, and it is not immediately clear how the results could be replicated. Moreover, model predictions are not validated - it is obvious that the model can infer something interesting (such as who is more less competitive), but not whether this actually resembles empirical observations. This is in my opinion a serious drawback of the paper.

Ideally, authors would i) describe the data in more detail; ii) include scripts they used to analyze the data – four parameters on individual level are not at all easy to estimate from limited data; authors mention parameter constraints in SI but how these were chosen and how they affect results is described very scarcely; iii) validate model predictions in some way, ideally by comparing them to an unused/additional part of the data; iv) if iii) is not possible with this data set, include and analyze another data set where this is possible.

With those changes, this could be a major contribution, deserving of publication in *Nature Communication*.

Reviewer #3:

Remarks to the Author:

The authors propose a new game-theoretic model of social network formation. Nodes are assumed to decide about the creation of their own ties (e.g., decide whom to follow on twitter or who to communicate with) based on a utility maximization strategy. The nodes' evaluation takes into account a number of criteria like the costs of ties and the benefits / costs of having a network position with high degree, reciprocal connections, and transitive embedded structures. This paper extends important bodies of literature that aim at i) providing rigorous mathematical models for network data, and ii) expressing network formation as a process that depends on a variety of social mechanisms or motifs. The paper is excellent in terms of its mathematical core. The authors present very nice proofs about the link between the behavioral model and the stable PNE/NE outcomes. The paper is well written. The embedding in social science theory, statistical network modeling, empirical network research are less elaborated and I provide a number of suggestions below.

I) Social science theory

I think the authors took a great effort in linking their mathematical model to social science theory. I think this is an important step. However, I partly feel that there are too many symbolic citations and too little engagement with the actual arguments in the papers cited. I have a few questions and suggestions on how to improve this part.

- Structural holes as introduced by Burt is a local concept involving actors in distance one from the focal actor. Betweenness centrality is defined on all paths of a graph. The authors are not wrong when they say that the concepts are related, but then it unclear how their measures are actually a good representation of structural holes motifs (see my comment below on the model).

- Heider's theory is not related to the "psychological principle of cohesion and support". Cohesion is correctly identified as an outcome, but Heider's argument about why transitive structures emerge builds upon Festinger's cognitive dissonance theory. It argues that imbalanced situation may cause stress for individuals and therefore the need to balance them out (by forming balanced triads or dissolving imbalanced triads). Besides the fact that both literature strands explain the formation of triads, social support and balance theory are very different. There are a number of additional explanations for triadic closure that could equally be considered, for example, transitivity as a result of homophily, social foci or spatial structures.

- Argue why it would be reasonable that twitter networks (given as an example) are the consequence of a strategic network formation. What about alternative and potentially unobserved explanations such as algorithms, cognitive biases, other types of randomness? This question relates to the lack of an error term in the model discussed below.

- I would suggest to similarly discuss some social science theory on reciprocity and cycle structures in networks (see my suggestion on the model below).

II) Link to inferential statistics

The authors claim that "with our parametric model, we are able to reverse the typical approach in the strategic network formation literature and infer individual incentives from stable network architectures." I would call this reversed approach inferential statistics and I would further suggest to

more thoroughly review the literature on inferential network methods (see, e.g., the textbook by Robins, 2015). There is in fact a few decades of work on how to infer individual incentives and motivation from stable network structures. Most relevant might be the work on exponential random graph models (see, e.g., Lusher et al., 2013). Given the fact that the authors assume that "each agent of the network has control on the weights of her outgoing links, while she cannot change her incoming links" a reference to and comparison with stochastic actor-oriented models (Snijders, 1996) might be useful as well.

What is the quality criterion of the achieved empirical results in terms of deviation from an optimal Nash equilibrium? Providing and calculating such a criterion would be important and similar to, for example, explained variance criteria in basic regression models. It could be used for model evaluation and comparison.

Is the main goal of the ML approach in fact to "recognizing and clustering similar individual behaviors"? This is a rather untypical case study, and if this is the main point, this should be highlighted more clearly. The clustering approach reminds me of latent statistical models and again it would be helpful to provide a brief comparison.

In the heterogeneous model, the goal is not anymore to prove under what parameters a (P)NE optimum can be found for a given network, but under which node parameters the model is getting as close as possible to an NE (see the comment above on a lack of a quality criterion). It is great that the ML estimation works. But in analogy to inferential statistics I would strongly suggest to provide some measure of confidence (similar to standard errors/ confidence intervals). A practical way how to approach this problem could be to re-estimate parameters with a network that is slightly perturbed. If node estimates are stable, this increases the confidence in the subsequent nodal clustering.

Why is there no error term in the statistical model? It seems to me that at least since the work of Arrow, Luce and others in the 60's random utility maximization models are considered a standard in strategic choice models; at least when empirical data is fitted. I guess the reasons are analytical (which is completely fine) but it should be explained why they opted for a no-random utility model?

III) Game-theoretical model

The authors consider that ties are costly (γ parameter). But could the model also consider that the change of ties could be costly as well, not just their maintenance? I understand that this might lead to some mathematical complications, but maybe the reasons for tie costs could at least be discussed briefly.

I think it should be clarified earlier on that the extended indegree measure ("influence") of function $t()$ among others also includes reciprocal structures (if you follow me, I follow you) and three-cycle structures. Both structures are central motifs in network models and should thus also appear in the introductory section. There is, for example, a broad body of literature in how far cycles are atypical structures in hierarchical networks (e.g., Davis 1970). The cycle structure is in the literature often contrasted to the transitivity motif discussed next. Introducing reciprocity explicitly will make it intuitively clear why, for example, the empty graph is only a PNE when the costs of tie formation are higher than the benefits of pairwise (= reciprocal) coordination.

It is argued that the β parameter can be negative and then be in line with Burt's theory on structural holes. It would then be less common that two paths a_{il} a_{lk} are closed by a direct connection a_{ik} . But isn't it individual l who is actually the broker in this model and thus the one who

has an increased utility by bridging a structural hole? Why would i have an incentive not to destroy l's brokerage position? Can you clarify how this fits to your actor-control approach?

V) Suggestions about the framing

The following two comments relate to the framing of the study. They are merely suggestions and not requests in terms of the forthcoming R&R.

The resulting networks are highly stylized (e.g., empty, complete, star graph). There is a strand in the network science literature (highly cited, less useful for practical purposes) in which similarly stylized network models have been proposed and fitted to data. Examples are preferential attachment models, stochastic blockmodels, small world models. It might be an idea to connect to the analysis to such models and explain whether and in how much they deviate from perfect NE outcomes.

Another literature that might be worth exploring to build upon is the work on model degeneracy (and the conditions of degeneracy) for exponential random graph models.

Statement of Revision

Comments by Reviewer # 1

[R1: 1] “ THE PAPER “SOCIAL NETWORK FORMATION: FROM SYSTEMIC STABILITY TO INDIVIDUAL INCENTIVES” STUDIES THE CREATION OF SOCIAL NETWORKS MAKING USE OF A MODEL BASED ON GAME THEORY. IN PARTICULAR, THE PAY-OFF OF THE DIFFERENT AGENTS IS A TRADE-OFF BETWEEN DIFFERENT TOPOLOGICAL QUANTITIES THAT DESCRIBE INFLUENCE, EFFICACY AND SAFETY, AND TWO DIFFERENT STABILITY DEFINITIONS ARE USED (NASH EQUILIBRIUM AND PAIRWISE-NASH EQUILIBRIUM). THE MODEL CONFIRMS SOME RESULTS ALREADY SEEN IN THE LITERATURE AND YIELDS SOME ORIGINAL CONCLUSIONS, AND IT IS USED TO DESCRIBE A REAL CASE. THE METHODOLOGY STUDIES WEIGHTED AND DIRECTED NETWORKS, WHICH IMPROVES ITS APPLICABILITY. THE PAPER IS WELL WRITTEN, IT IS CLEARLY EXPLAINED, THE AUTHORS HAVE DEVELOPED EXTENSIVE ANALYTICAL AND NUMERICAL WORK, AND THE SUBJECT UNDER STUDY COULD BE OF HIGH INTEREST FOR RESEARCHERS WORKING ON SOCIAL AND ECONOMIC NETWORKS. THE FIGURES ARE CORRECTLY DONE. ”

We greatly thank Reviewer #1 for her/his comments, careful reading, and positive evaluation of our work.

[R1: 2] “ WHILE I BELIEVE IN THE POTENTIAL GENERAL INTEREST OF THIS PAPER, MY MAIN COMPLAINT IS THAT IN THE PRESENT MANUSCRIPT THE WIDE IMPACT OF THE RESULTS IS NOT SUFFICIENTLY DESCRIBED. FOR THIS REASON, SOME OF MY MAIN COMMENTS ARE DIRECTED TO ENLARGE THE POTENTIAL INFLUENCE AND IMPACT OF THE WORK IN THE COMMUNITY. IN PARTICULAR, SOCIOLOGY, ECONOMICS AND NETWORK THEORY RESULTS HAVE VERY OFTEN BEEN VERY DISCONNECTED FROM EACH OTHER, WITH NO INTERACTION AMONG THE THREE DIFFERENT COMMUNITIES. IN FACT, TOO MANY TIMES THE SAME RESULTS HAVE BEEN FOUND AND DEVELOPED IN PARALLEL IN THESE THREE DIFFERENT CONTEXTS. AS NATURE COMMUNICATIONS IS DEVOTED TO A WIDE VARIETY OF POTENTIAL READERS, THIS IS A VERY GOOD OPPORTUNITY TO COLLABORATE TO SOLVE THIS ENDEMIC PROBLEM OF THE FIELD. ATTENDING TO THIS SEARCH FOR GENERALIZATION, THERE ARE SOME MAIN QUESTIONS AND SOME OTHER MINOR DETAILS THAT SHOULD BE CLARIFIED OR IMPROVED BEFORE I CAN RECOMMEND THE PUBLICATION OF THIS PAPER IN NATURE COMMUNICATIONS. ”

We agree with the reviewer that such a generalization would be extremely beneficial to reach a much wider audience and to fill the long-standing gap between these different communities. In this regard, we improved the revised version of the manuscript by addressing three types of generalizations:

- we introduced a new section called “Random Networks” where we use our behavior estimation method to give a sociological and strategic interpretation of the probabilistic rules behind two well known examples of random networks, namely the Preferential Attachment and the Small-world models (see [R1:3,7] and [R3:6(a)]),
- we improved our estimation method by means of a rigorous statistical analysis, creating a connection with Stochastic Actor Oriented Models (see [R3:4]),
- we extended the analysis of real-world networks to meet the standard requirements in the empirical literature, e.g., the validation process ([R2:3(c),(d)]).

All of them were triggered by constructive comments of the reviewers and we believe that our efforts to connect the different approaches also make the paper attractive to different communities. We further elaborate on them below.

[R1:3] “MY FIRST COMPLAINT IS THAT THE PAPER, ESPECIALLY IN THE INTRODUCTION, HAS FOCUSED ON THE WORK DEVELOPED ON THE SOCIOLOGY AND ECONOMICS LITERATURE (MAINLY IN THE LATTER), BUT HAS LARGELY IGNORED MOST WORK DONE IN THE CONTEXT OF COMPLEX NETWORKS, WHICH IS MORE GENERAL AND IS ALSO EXTENSIVE IN THE FIELD (CASTELLANO ET AL., 2009). I ADDRESS THE AUTHORS TO START INCLUDING (AT LEAST IN THE INTRODUCTION) A DETAILED STATE OF THE ART OF SOCIAL NETWORK FORMATION FROM THE GENERAL PERSPECTIVE OF NETWORK SCIENCE/COMPLEXITY. IN FACT, PLEASE NOTE THAT THE LITERATURE CITED IN THE SOCIO-ECONOMICS CONTEXT IN THE PAPER IS RELATIVELY OLD (MAINLY BEFORE 2012), WHICH COULD SHOW THAT THE SUBJECT IS LOSING IMPORTANCE IN THE LAST DECADE. A THOROUGH ANALYSIS OF MORE GENERAL LITERATURE WILL SHOW THAT THIS IS NOT THE CASE, AND THAT THE FIELD IS QUITE “ALIVE” IN THE NETWORK SCIENCE CONTEXT. ”

We agree with the reviewer that such a wider literature review can significantly improve the quality of the manuscript in terms of visibility, applicability and opportunities of comparisons. Thus, we thorough revised the literature review in the introduction, which now covers famous random networks models from complex networks, e.g., ER, small-world, and preferential attachment. In a similar respect, we also included a link to inferential statistics, and in particular to Stochastic Actor Oriented Models (SAOM) and Exponential Random Graph Models (ERGM) as suggested by Reviewer #3 (see [R3:4]). Below, we report our change in the introduction:

“Starting from the random graph model proposed by Erdős and Rényi[13, 14], the complex networks community proposed a number of network formation models driven by sociological observations and supported by empirical evidence. Among them, the small world network model introduced by Watts and Strogatz [35] shows that the addition of few random ties to a regular lattice (highly locally connected) results into a small diameter network, as in Milgram’s experiment [25] on the six degrees of separation. To explain the emergence of scaling in random networks, Barabási and Albert proposed the preferential attachment model [2], in which newborn nodes select their connections proportional to popularity. A broad literature on complex (social) networks and dynamics thereof has grown ever since (see [8, 21, 10]). While such probabilistic models can successfully reproduce the macroscopic statistical structural properties of social networks, they do not offer insights into the sociological microscopic foundations.

A different socio-theoretical and statistical approach was proposed by Snijders with Stochastic Actor-Oriented Models[31] (SAOM). Based on the idea that nodes of the graph are social actors having the potential to change their outgoing ties, the observed network is the result of the actors’ behavior[32]. The preference or payoff function that each actor tries to maximize is split into a modeled and a random component, where the modeled component contains statistical parameters that have to be estimated from the available data through likelihood-based methods [33]. As in generalized linear statistical models, the objective function is assumed to be a linear combination of a set of components, called effects, e.g., reciprocity, transitivity, or the tendency of having ties at all [33]. Similarly to SAOM, Exponential random graph models (ERGM)[22] study network configurations, which are small subsets of possible network ties (and/or actor attributes), e.g., reciprocated ties[30]. Yet, the focus is on ties rather than on actors. ”

On the other hand, as Castellano et al. say[10], statistical physics of social dynamics attempt to understand “regularities at large scale as collective effects of the interaction among single individuals. [...] With this concept of universality in mind, one can approach the modelization of social systems, trying to include only the simplest and most important properties of single individuals and looking for qualitative features exhibited by models.” The contribution of the statistical physics community spans from opinion and crowd dynamics to social and epidemics spreading [28], to name but a few. However, to the best of our knowledge, there is no specific literature on network formation processes in the statistical physics, as the attention is put on “high level features, such as symmetries, dimensionality, or conservations laws, [...] relevant for the global behavior”[10], rather than on the detailed complex behavior of the individuals. Despite socio-physics and socio-dynamics being very exciting fields, we believe that they are not fully aligned with the scope of our research. Thus, aside from a few references in the opening paragraphs (see above), we decided to limit the detailed literature review to the considered random network models, Stochastic Actor-Oriented Models, Exponential random

graph models, as well as the literature on strategic network formation, which is most aligned with our paper.

[R1:4] “I AGREE WITH THE AUTHORS IN THE FACT THAT THERE IS STILL NO COMMON AGREEMENT ON THE TYPE OF CENTRALITY THAT SHOULD BE USED WHEN STUDYING THE EVOLUTION / CREATION OF NETWORKS. HOWEVER, IN THE LAST DECADE SOME QUANTITIES NOT CITED IN THE PAPER HAVE RESULTED TO BE VERY USEFUL IN THIS CONTEXT (ANOTHER CONSEQUENCE PERHAPS OF THE DISCONNECTION BETWEEN DISCIPLINES MENTIONED ABOVE). JUST AN EXAMPLE: THE EIGENVECTOR CENTRALITY HAS BEEN WIDELY USED TO MEASURE KNOWLEDGE ASSOCIATED TO MODELS OF INNOVATION AND CULTURE SPREADING (KÖNIG ET AL, 2008), THE PROBABILITY OF BEING INFECTED IN EPIDEMIOLOGY (NEWMAN, 2010), OR THE IMPORTANCE OF WEBPAGES (LANGVILLE, 2006). IT HAS ALSO BEEN RECENTLY APPLIED IN THE GAME THEORY CONTEXT IN THIS JOURNAL (IRANZO ET AL., 2016), WHERE SEVERAL NETWORKS COMPETE FOR CENTRALITY ASSOCIATED TO NASH EQUILIBRIA. FURTHERMORE, THERE IS ALREADY EXPERIMENTAL PROOF OF THE PREVALENCE OF THE EIGENVECTOR CENTRALITY OVER OTHER TOPOLOGICAL MEASURES IN REAL SYSTEMS: IN (BANERJEE, 2013) MICROFINANCE PARTICIPATION IN RURAL INDIAN VILLAGES WAS SHOWN TO BE SIGNIFICANTLY HIGHER WHEN THE INDIVIDUALS USED AS INJECTION POINTS SHOWED HIGHER EIGENVECTOR CENTRALITY. IN SUMMARY, PLEASE COMPARE THE RESULTS OBTAINED BY USING THE NOVEL DEFINITION OF THE PAY-OFF PRESENTED IN THIS PAPER WITH A MORE GENERAL DESCRIPTION OF THE FIELD. ”

We thank the reviewer for her/his comment. Certainly, we agree with her/him that other measures, e.g., eigenvector centrality, have received attention in the network formation literature. As a matter of fact, in a previous version of our model we considered eigenvector centrality as main driving force in the payoff function. The results, not yet submitted, suffer a major drawback deriving from the fact that the eigenvector centrality is hard to perceive by individuals within a network due to its intrinsically non-local definition. The same comment applies, in fact, to Katz centrality, as emphasized in the paper. Hence, in our model, we prioritize locally-assessable measures which are compatible with the agents’ limited information assumption. For this reason, we opted for the truncated Katz centrality as a locally assessable metric.

Another independent reason why we pursued the truncated Katz centrality here, are the immediate connections with Stochastic Actor models from the sociology literature; see our response to comment [R3:4] and the new section “Socio-theoretical interpretation”. In order to emphasize our focus on locally-assessable measures with strong sociological interpretation, we introduced the following modification in the revised version of the manuscript:

“Such a measure [influence] extends the indegree centrality definition by introducing the contribution of the strength of all weighted paths of length 2 and 3 which are ending in i , discounted with factors δ_i and δ_i^2 [. . .]. This measure can also be viewed as an approximated Katz centrality. In the original definition, Katz [17] considers paths of all lengths, yet in real-world social networks agents have limited information on the network topology (one can think of LinkedIn’s 3rd degree of separation). Compared to that, our definition is locally-assessable, i.e., it does not require complete information of the entire network, yet it includes most important social networks patterns, such as diads and triads [30].”

Furthermore, we agree with the reviewer that eigenvector centrality has been proved to be a powerful tool to analyse diffusion processes of innovation and culture spreading [19], in epidemiology [26], and even from an empirical point of view [1]. However, the dynamics underlying these diffusion processes differ from the social network formation dynamics, which are rather driven by sociological incentives such as reciprocity, clustering, transitivity, homophily, to name but a few. Similarly, in [20] and [16], the nodes under consideration are web-pages and villages, whose dynamics do not fall into the same category. Nevertheless, we agree with the reviewer that it is worth mentioning it in the conclusion as future direction: *“We emphasize that our model can be adapted to different descriptions of the payoff function, e.g., considering an extra cost for changing ties, or other individual incentives such as eigenvector centrality, or constraining competitors’*

brokerage [...]”.

[R1:5] “ONE OF THE MAIN TARGETS OF THE PAPER IS TO SHOW THAT COMBINING MORE THAN ONE TOPOLOGICAL QUANTITY IN THE PAY-OFF OF THE NODES WHEN DESCRIBING STRATEGIC NETWORK FORMATION CAN REFLECT THE REAL PHENOMENOLOGY MORE ACCURATELY THAN WITH A SINGLE ONE. IN MY OPINION THIS IS A SUGGESTIVE IDEA THAT DESERVES THE THOROUGH ANALYSIS THAT THIS PAPER DEVELOPS. HOWEVER, AND AS THE AUTHORS AFFIRM, A SIMILAR QUESTION WAS ALREADY DESCRIBED IN [3], WHERE BOTH BETWEENNESS AND CLOSENESS WERE COMBINED. ALSO, IN (GRAUWIN ET AL, 2009) A PAY-OFF WAS INTRODUCED AS A CONTINUOUS INTERPOLATION BETWEEN COOPERATIVE AND INDIVIDUAL DYNAMICS. THEN, WHICH ARE THE MAIN ADVANTAGES OF THIS NEW METHODOLOGY IN COMPARISON TO THE ALREADY EXISTING ONES? ”

We thank the reviewer for her/his comment and positive evaluation of our work. Indeed, our parametric approach aims at giving a unified expression to several distinct observations in the field of strategic network formation. “*The main limitation of these models lies in the isolation of specific centrality metrics, which prevents from a comprehensive analysis of the network topology stability with respect to multiple co-existing incentives.*” As specified in our article and as mentioned by the reviewer above, a first attempt to overcome this limitation was proposed in [3] with a parametric combination of closeness and betweenness centrality.

Our model offers several steps forward. Firstly, our payoff function also considers the effect of clustering, which is known to be extremely relevant in the social network context. Secondly, our payoff function provides a wider spectrum of combinations of parameters and incentives compared to the one in [3]. Note, in fact, that their payoff function is the result of a linear combination of betweenness and closeness, thus the different incentives cannot be simultaneously present, nor absent. Furthermore, we provide not only strong analytical results to the more general case of directed and weighted networks, but we also use our parametric formulation to fill the gap between theoretical, empirical and simulation-based results by means of our behavior estimation method.

Finally, we would like to thank the reviewer for pointing out an interesting direction, namely the analysis of a continuous interpolation between individual and collective dynamics, as in [15]. Following the network formation literature, in fact, we focused on two notions of equilibria: Nash and pairwise-Nash equilibria. The pairwise-Nash equilibrium already takes into account both selfish and cooperative behavior, however it does not offer the possibility of a continuous interpolation between these two factors, as proposed in [15]. We leave this interesting direction, which eventually requires a novel equilibrium definition, for future investigation, as we think it might burden the current analysis. Yet, we added the following comment in the conclusions:

“We emphasize that our model can be adapted [...] to different definitions of equilibrium, e.g., mixed-Nash equilibria or continuous parametric transitions of selfish-cooperative behavior as in [15].”

[R1:6] “TWO TYPES OF EQUILIBRIA ARE STUDIED, NASH EQUILIBRIUM AND PAIRWISE-NASH EQUILIBRIUM. THIS IS AN INTERESTING WAY TO MEASURE PURELY SELFISH AND PARTIALLY COORDINATED BEHAVIORS IN THE AGENTS. HOWEVER, I WOULD APPRECIATE SOME MORE INFORMATION RELATED TO THE ACTIONS THAT EACH NODE IS ALLOWED TO DO IN ORDER TO CHARACTERIZE MORE PRECISELY THE NASH EQUILIBRIA. IN PARTICULAR, CAN EACH NODE CHANGE THE WEIGHT OF ANY OF ITS OUTGOING LINKS AN ARBITRARILY LARGE OR SMALL QUANTITY, CREATE LINKS WITH ANY WEIGHT, ETC? IF THIS IS THE CASE, THEN THE SYSTEM WOULD FACE MIXED NASH EQUILIBRIA, WHICH ENABLES A PROBABILISTIC ANALYSIS OF THE SYSTEM AND INCREASES ITS APPLICABILITY. PLEASE DESCRIBE THIS QUESTION WITH SOME MORE DETAIL. ”

Thank you for this important comment. Yes, each node can change the weight of any of its outgoing links an arbitrarily large or small quantity, create links with any weight. Let us elaborate below.

When we consider the Nash equilibrium setting, agents are allowed to simultaneously change the weight of all their outgoing ties. This emerges in the Nash equilibrium definition, which we report here:

Definition. \mathcal{G}^* is a Nash equilibrium (NE) if

C1. for all agents i , $V_i(a_i, \mathbf{a}_{-i}^*) \leq V_i(a_i^*, \mathbf{a}_{-i}^*)$, $\forall a_i \in \mathcal{A}$.

“Note that agents are allowed to play any action in the space \mathcal{A} , i.e., to simultaneously change all the outgoing ties.” We also report the definition of the action space: “In game theoretical language, a typical action of agent i can be represented as

$$a_i = [a_{i1}, \dots, a_{i,i-1}, 0, a_{i,i+1}, \dots, a_{iN}],$$

living in the action space $\mathcal{A} = [0, 1]^{N-1}$. Conversely, we denote a typical action of all agents but i as

$$\mathbf{a}_{-i} = [a_1; \dots; a_{i-1}; a_{i+1}; \dots; a_N] \in \mathcal{A}^{N-1}."$$

For completeness, we also emphasize that in the Pairwise-Nash equilibrium setting, deviations are only allowed by pairs of agents (i, j) , and are restricted to the mutual ties a_{ij} and a_{ji} . All this information are accessible to the reader and should be even more clear in the proofs provided in the SI.

We agree with the reviewer that mixed Nash equilibrium analysis might apply to our model. On the other hand, to the best of our knowledge it has never been applied in the context of strategic network formation. Furthermore, being our network directed, our action space continuous, and our focus on observed networks (mixed Nash equilibrium are usually related to repeated games), we believe that such a generalization might not be so relevant at this stage of our research and we believe it is outside the scope of this article. However, we agree on the importance of studying different equilibrium concepts, thus we have explicitly mentioned this in the conclusions as an avenue for future research:

“We emphasize that our model can be adapted [...] to different definitions of equilibrium, e.g., mixed-Nash equilibria [...].”

[R1:7] “IN THE MANUSCRIPT, FOUR VERY BASIC TYPES OF NETWORKS WERE ANALYTICALLY STUDIED. THIS IS INTERESTING BECAUSE THEORETICAL WORK IS A POWERFUL TOOL, BUT WHILE MOST WORK ON ECONOMIC MODELS ARE RESTRICTED TO CLIQUES AND STARS OF DIFFERENT SIZES, IT IS KNOWN THAT MOST REAL R&D NETWORKS ARE SPARSE, LOCALLY DENSE AND SHOW HETEROGENEOUS DEGREE DISTRIBUTIONS (COWAN, 2004; POWELL ET AL., 2005). IN GENERAL, NETWORKS WITH DIFFERENT TOPOLOGY BEHAVE VERY DIFFERENTLY. I WOULD APPRECIATE A GENERALIZATION (PERHAPS ONLY NUMERICAL) TO MORE REALISTIC NETWORKS (SCALE-FREE NETWORKS, RANDOM ER, SMALL-WORLD, OR REGULAR NETWORKS, TO CITE JUST A FEW) TO SHOW THE REAL APPLICABILITY OF THE METHODOLOGY. THIS WOULD INCREASE THE IMPACT OF THE PAPER SUBSTANTIALLY, AND IT WOULD BECOME ATTRACTIVE FOR SCIENTISTS BEYOND THE ECONOMIC COMMUNITY. ”

We greatly thank the reviewer for suggesting this promising direction, which gave us the opportunity of exploring a closely related field. As pointed out by the reviewer, theoretical analysis is a powerful tool, yet it is often restricted to stylized models and detached from empirical studies. Complete, star, bipartite and empty networks can be the result of a rational strategic process, but they rarely constitute examples of real-world networks. Conversely, the complex networks community developed a number of random networks models whose aim is to reproduce the common features of real-world networks. At the same time, it is hard to immediately and explicitly relate such models to a socio-economic strategic behavior of the agents. Furthermore, the probabilistic rules typically define the behavior at an aggregate level, rather than at the single individual level.

According to these observations, the gap between these two approaches offers an interesting opportunity. Thus, we revised our manuscript by refining our behavior estimation method (see our answer to [R3:4]) and introducing a new section called “Random Networks” whose aim is to use this method to give a sociological and strategic interpretation of the probabilistic rules behind two famous examples of random networks, namely the Preferential Attachment and the Small-World models. Our numerical results offer a

novel explanation of these models in terms of sociological features such as reciprocity, clustering, structural holes and cyclic structures, laying the ground for a potentially promising connection between the two fields. Please, see [R3:6(a)], [R3:4], [R3:5(b)] for related comments, and the new section called “Random Networks” for further details.

[R1: 8] “IN SEVERAL FIGURES THERE ARE PARAMETER REGIONS WHERE DIFFERENT NASH EQUILIBRIA CO-EXIST. WOULD IT BE POSSIBLE TO STUDY THE TRANSITIONS FROM ONE EQUILIBRIUM TO THE REST? THAT IS, IF ONE NODE DECIDED TO CHANGE ITS CONNECTION (OBVIOUSLY LOSING SOME PAY-OFF), WOULD THE REST BE OBLIGED TO CHANGE ALSO THEIR CONNECTIONS AND PUSH THE SYSTEM TOWARDS A DIFFERENT EQUILIBRIUM? THIS ANALYSIS WOULD YIELD VALUABLE INFORMATION ABOUT THE SYSTEM, SUCH AS WHETHER SOME EQUILIBRIA ARE MORE STABLE THAN THE REST (IRANZO ET AL, 2016). ”

The reviewer’s comment points towards an interesting direction. Studying the robustness of the equilibria against possible perturbations, whether endogenous or exogenous, or their region of attraction under (e.g., best-response) dynamics, constitutes by itself an interesting research question. Understanding and modelling cascade effects that can bring a complete network to its opposite, the empty network, or that can turn a star (dictatorship) into a complete network (democracy) are certainly directions worth to explore. In the paper, we have already partially addressed this question when studying the Nash equilibrium condition of the complete network motif: “Concerning the NE, if the ties are too costly ($\gamma > \bar{\gamma}_{NE}$), the best action for each agent is to drop all outgoing ties. This transition behaviour has dramatic consequences, as it leads to the empty network if agents simultaneously play a best response.”

Despite the relevance of this investigation, we decided to leave it for future work in favor of other extensions (see points (i)-(iii) in [R1:2]) that are more focused towards a validation, comparison, and application of our model. However, we explicitly mention the suggested extension in the conclusions as a promising avenue for future work:

“We confirmed existing results on the correlation between incentives and stable network architectures, yet revealing new transition paradigms and opening the door for future investigation on cascade effects and robustness of equilibria.”

[R1: 9] “ IN SUMMARY, IF THE AUTHORS ADDRESS MY COMMENTS AND COMPLAINTS WITH THE AIM OF INCREASING THE IMPACT AND APPLICABILITY OF THE RESULTS PRESENTED IN THEIR MANUSCRIPT I WILL BE HAPPY TO RECOMMEND THE PAPER FOR PUBLICATION IN NATURE COMMUNICATIONS. ”

We thank the reviewer for all the constructive suggestions and the positive evaluation of our work.

[R1: 10] “MINOR COMMENTS: ”

- “IN THE ABSTRACT THE OUT-OF-EQUILIBRIUM DYNAMICS OF THE SYSTEM IS CITED, BUT AS FAR AS I HAVE READ THE WORK FOCUSES ON NASH EQUILIBRIA. PLEASE CLARIFY THIS QUESTION.”

We agree with the reviewer that the sentence in the abstract did not have a clear follow-up in the paper. Even though our analytical part focuses on equilibria, the proofs (available in the SI), suggest that our necessary and sufficient conditions are closely related to the dynamics that would appear as soon as these conditions are not satisfied. To give an example, when we consider the star network, we identify three conditions for the stability, namely: “(i) the central node must have no incentive in dropping her ties, (ii) the periphery nodes must not destroy the link to the center of the star, and (iii) must not initiate ties among them”. The parametric conditions proved in the theorem precisely aim at meeting these requirements. Thus, when one of this parametric condition is not fulfilled, for instance when $\gamma > \alpha\delta(1 + \delta)$, the out-of-equilibrium dynamics are suggested by (ii), namely the periphery nodes have an incentive in

dropping the tie towards the center. This, and similar observations can be evinced by the proofs. Nonetheless, we agree with the reviewer that the focus of the article is not on the out-of-equilibria dynamics, thus we diminished the attention drawn on the topic by removing it from the abstract, the introduction and the conclusions.

- “IN FIG. 5 THE TEXT AFFIRMS THAT THERE ARE COEXISTENCE OF ALL FOUR TYPES, BUT I CAN’T SEE IT IN THE FIGURE.”

In fact, the coexistence can be identified only in a small region in Fig. 5. We enlarged the figure in the revised version of the manuscript to make it more clear.

- “IN FIG. 7 THE AUSTRALIAN BANK DATA SET WAS ANALYZED AS A NASH EQUILIBRIUM. WHY SHOULD THE SYSTEM HAVE ALREADY REACHED SUCH EQUILIBRIUM? PLEASE EXPLAIN IN MORE DETAIL. ”

We thank the reviewer for this observation. As a matter of fact, we *assume* the data collected within a specific snapshot constitutes an equilibrium. As this might not be entirely true, we revised our behavior estimation method introducing an error term, accounting for bounded rationality of the agents, as well as for noisy observations. See the answer to [R3:4] (in particular, [R3:4(a), (d)]), for more details on the method and on the error term.

In order to clarify on this, we emphasized that the data observed constitutes “approximately” a Nash equilibrium, specifically adding the following sentence in the SI:

“In our behavior estimation method, we consider agents with heterogeneous individual preferences’ sets $P_i = \{\alpha_i, \beta_i, \gamma_i, \delta_i\}$. We assume to observe a network \mathcal{G}^ of N agents, where the actions a_i^* of the agents are “approximately” a Nash equilibrium with respect to the payoff functions $V_i(a_i, \mathbf{a}_{-i}, P_i)$ which depends on some unknown parameters P_i .”*

- “WHEN DESCRIBING NASH EQUILIBRIUM (NE), THE AUTHORS CITE LINKEDIN CONNECTIONS AS AN EXAMPLE. HOWEVER, IN LINKEDIN EVERY TWO NODES CONNECTED BY A LINK MUST AGREE TO CREATE SUCH LINK WHILE IN THE NE OF THE PAPER EACH AGENT TOTALLY CONTROLS ITS OWN OUTGOING LINKS. PLEASE CLARIFY.”

We agree with the reviewer that this example is imprecise. We changed this part accordingly, and now it reads as:

“This is a reasonable approach in many competitive contexts or marketing environments, e.g., when agents strategically retweet or choose their Instagram followees.”

- “PLEASE DEFINE PARETO OPTIMALITY CONDITION AND PARETO OPTIMAL FRONT FOR COMPLETENESS.”

We thank the reviewer for the comment. We added the definition in the SI and a reference to it in the main body.

“Let us first review the definition of Pareto optimality in economics (see [23]).

Definition. Consider an economy with n agents and k goods. Then an allocation $x = \{x_1, \dots, x_n\}$, where $x_i \in \mathbb{R}^k$ for all i , is Pareto optimal if there is no other feasible allocation $\{x'_1, \dots, x'_n\}$ such that, for utility function u_i for each agent i , $u_i(x'_i) \geq u_i(x_i)$ for all $i \in \{1, \dots, n\}$ with $u_i(x'_i) > u_i(x_i)$ for some i . Moreover, the set of all Pareto optimal allocations constitutes the Pareto optimal front.

In other words, the Pareto optimality condition requires that there exists no allocation of goods which strictly increases the payoff of at least one agent while not decreasing the payoff of the others.

On the other hand, Condition C 3 of the pairwise-Nash equilibrium requires that for all pairs (i, j) , and for all pairs (a_{ij}, a_{ji}) in $[0, 1]^2$,

$$\begin{aligned} V_i(a_{ij}, a_{ji}, \mathbf{a}_{-(i,j)}^*) &> V_i(a_{ij}^*, a_{ji}^*, \mathbf{a}_{-(i,j)}^*) \\ \Downarrow \\ V_j(a_{ij}, a_{ji}, \mathbf{a}_{-(i,j)}^*) &< V_j(a_{ij}^*, a_{ji}^*, \mathbf{a}_{-(i,j)}^*). \end{aligned}$$

In other words, it is satisfied if there exists no other pair (a_{ij}, a_{ji}) in $[0, 1]^2$ such that i and j are simultaneously better off, with at least one of the two being strictly better off. Thus, Condition C 3 in fact corresponds to a Pareto Optimality condition. "

We finally thank the reviewer again for all her/his constructive comments which prompted a lot of changes in the paper and led to an improved manuscript.

Comments by Reviewer # 2

[R2: 1] “I REALLY LIKE THE FRAMEWORK DESCRIBED IN THIS PAPER, WHICH ENABLES STUDYING AND COMPARING A NUMBER OF POSSIBLE INCENTIVES THAT ACTORS CAN HAVE WHEN FORMING THEIR NETWORKS. BESIDES ITS THEORETICAL ELEGANCE, A MAJOR BENEFIT OF THE FRAMEWORK IS THAT IT HAS THE POTENTIAL TO BE USED TO INFER ACTUAL INDIVIDUAL INCENTIVES FROM REAL-WORLD STABLE NETWORK ARCHITECTURES. THIS CAN BE VERY VALUABLE, AS MANY ELEGANT MODELS CAN BE DEvised BUT NOT MANY ARE ACTUALLY USEFUL. ”

We greatly thank Reviewer #2 for her/his comments, careful reading, and positive evaluation of our work.

[R2: 2] “ HOWEVER, THE EMPIRICAL PART OF THE PAPER IS CURRENTLY SOMEWHAT GLOSSED OVER. THE DATA SET USED IS SCARCELY DESCRIBED, AND IT IS NOT IMMEDIATELY CLEAR HOW THE RESULTS COULD BE REPLICATED. MOREOVER, MODEL PREDICTIONS ARE NOT VALIDATED - IT IS OBVIOUS THAT THE MODEL CAN INFER SOMETHING INTERESTING (SUCH AS WHO IS MORE LESS COMPETITIVE), BUT NOT WHETHER THIS ACTUALLY RESEMBLES EMPIRICAL OBSERVATIONS. THIS IS IN MY OPINION A SERIOUS DRAWBACK OF THE PAPER. ”

We agree with the reviewer that not enough attention was devoted to the empirical part. Accordingly, the current revision of our manuscript addresses this issue on two aspects: (i) additional analysis of a widely studied data set (Florentine families) and deeper study of the Australian bank data set already included in the previous version; (ii) in-depth description of the behavior estimation method. Please find our detailed answer below.

[R2: 3] “IDEALLY, AUTHORS WOULD”

(a) “DESCRIBE THE DATA IN MORE DETAIL;”

As suggested by the reviewer, we provided further details on the Australian bank data set. More emphasis has been earmarked to the description of the data collection, including the background of the study, the question from which the network is constructed, namely “*In whom do you feel you would be able to confide if a problem arose that you did not want everyone to know about?*”, and some useful insights to address the model predictions validations (see below). Furthermore, we applied the same constructive suggestion to the newly inserted analysis of the Medici data set by describing the derivation of the directed settings (this data set is more frequently analyzed in its undirected version) and emphasizing many historical and socio-economical observations from the original paper [27].

(b) “INCLUDE SCRIPTS THEY USED TO ANALYZE THE DATA – FOUR PARAMETERS ON INDIVIDUAL LEVEL ARE NOT ALL EASY TO ESTIMATE FROM LIMITED DATA; AUTHORS MENTION PARAMETER CONSTRAINTS IN SI BUT HOW THESE WERE CHOSEN AND HOW THEY AFFECT RESULTS IS DESCRIBED VERY SCARCELY;”

The scripts we used to analyzed the data, already available in the previous version, have been updated and commented in more detail. We also added the following comment in the SI:

“*To conclude, note that the code that performs the behavior estimation method as well as the data sets discussed and the tests of the random network models are available at the following link:*

https://git.ee.ethz.ch/pagann/learning_strategic_behavior”.

Moreover, the behavior estimation method has been improved in this revised version, according to the suggestion of Reviewer #3. Estimates of the parameters now follow from a gradient method which does not require gridding. This allowed us not only to remove the parameter constraints, but also to perform a rigorous statistical analysis and to derive confidence intervals on our estimates. For more details, please see the Methods section, the SI and the answer to [R3:4].

- (c) “ VALIDATE MODEL PREDICTIONS IN SOME WAY, IDEALLY BY COMPARING THEM TO AN UNUSED / ADDITIONAL PART OF THE DATA;”

We thank the reviewer for such an interesting suggestion. Concerning the Australian Bank data set, we could not find support of all our predictions into the original paper [29], nor we could compare to other strategic network formation studies of the same data set. Yet, we attempted the task by matching some of our conclusions with the observations made by Pattison et al. [29]. For instance, we included the following sentence:

“From the analysis one evinces that more competitive behaviors (negative values of $\hat{\theta}_3$) are typical of high hierarchical positions, e.g., Branch and Deputy manager. Conversely, low-ranking positions are more inclined towards social support (positive $\hat{\theta}_3$), as witnessed by the behavior of tellers 1-6. As observed by Pattison[29], confiding relations are likely to be more local or restricted in their span, linking individuals from one level in the organization to those in the next. Thus, it is unlikely that high-rank agents exhibit clustering behavior, as there are fewer nodes in the top level of the hierarchical tree structure.”

Moreover, thanks to Reviewer #3 (see [R3:3(d)] and [R3.5(b)]), we highlighted the lack of cyclic structures, which emerges from our behavior estimation analysis and is known to be typical of hierarchical networks. In the revised manuscript we added the following comment:

“The complete analysis reported in the SI also shows that agents are not particularly inclined towards cyclic structures, in accordance with Davis[12] who showed that cycles are atypical structures in hierarchical networks.”

- (d) “ IF (C) IS NOT POSSIBLE WITH THIS DATA SET, INCLUDE AND ANALYZE ANOTHER DATA SET WHERE THIS IS POSSIBLE.”

We thank the reviewer for this alternative constructive suggestion, which stimulated an interesting journey through the analysis of other well-known data sets. Essentially, the reviewer is asking for a ground-truth example for which our inference (via the behavior estimation method) can be validated. We finally decided to perform our estimation analysis to the famous data set describing the marriage and business relationships among elite families in Renaissance Florence, originally collected by Kent [18], but first coded by Padgett and Ansell [27]. We focus our attention on the family of the Medici and the reason for this is twofold. Firstly because understanding its rise in power triggered a large interest in different communities outside the medieval history one, from sociology and economics to graph theory. Nonetheless, a large part of the literature on this example merely uses it to show a possible application of their model. Secondly, and most importantly, because the behavior of the Medici has also been extensively studied through historical and socio-economical interpretation. In other words, it is one of the very few data sets for which a “ground-truth” comparison is conceivable. As we were primarily interested in validating our model, we focused on the comparison of the results of our behavior estimation model with well-grounded historical observations proposed in the original paper by Padgett and Ansell [27].

In our revised version of the manuscript we have been able to illustrate a number of matchings between the historical data and our analysis of the strategic behavior of the Medici (see the “Medici network” section). To give some examples, our model captures the structural isolation operated by the Medici family showing their tendency towards a brokerage position. This behavior is consistent with the geographical and historical analysis carried on by Padgett and Ansell. Yet, our model shows that *“the structural isolation operated on multiplex ties not only guaranteed stability (preventing dissent spreading) but at the same time enhanced social (and political) support [...] to the Medici family”*. Furthermore, the analysis of our results on the reciprocity aspect confirms the theory of the segregation of types of ties supported by Padgett and Ansell. We invite the reviewer to read the entire Section on “Inference of Behavior for Complex Networks” for the details.

To conclude, even though the Australian bank data set did not offer a lot of opportunities for validat-

ing our model, we believe that the revised manuscript, together with the addition of the Medici data set, provides significant improvements in this respect.

[R2: 4] “WITH THOSE CHANGES, THIS COULD BE A MAJOR CONTRIBUTION, DESERVING OF PUBLICATION IN NATURE COMMUNICATION.”

We finally thank the reviewer for the positive evaluation and for all her/his constructive comments which led to an improved manuscript.

Comments by Reviewer # 3

[R3: 1] “THE AUTHORS PROPOSE A NEW GAME-THEORETIC MODEL OF SOCIAL NETWORK FORMATION. NODES ARE ASSUMED TO DECIDE ABOUT THE CREATION OF THEIR OWN TIES (E.G., DECIDE WHOM TO FOLLOW ON TWITTER OR WHO TO COMMUNICATE WITH) BASED ON A UTILITY MAXIMIZATION STRATEGY. THE NODES’ EVALUATION TAKES INTO ACCOUNT A NUMBER OF CRITERIA LIKE THE COSTS OF TIES AND THE BENEFITS / COSTS OF HAVING A NETWORK POSITION WITH HIGH DEGREE, RECIPROCAL CONNECTIONS, AND TRANSITIVE EMBEDDED STRUCTURES. THIS PAPER EXTENDS IMPORTANT BODIES OF LITERATURE THAT AIM AT I) PROVIDING RIGOROUS MATHEMATICAL MODELS FOR NETWORK DATA, AND II) EXPRESSING NETWORK FORMATION AS A PROCESS THAT DEPENDS ON A VARIETY OF SOCIAL MECHANISMS OR MOTIFS. THE PAPER IS EXCELLENT IN TERMS OF ITS MATHEMATICAL CORE. THE AUTHORS PRESENT VERY NICE PROOFS ABOUT THE LINK BETWEEN THE BEHAVIORAL MODEL AND THE STABLE PNE/NE OUTCOMES. THE PAPER IS WELL WRITTEN.”

We greatly thank Reviewer #3 for her/his comments, careful reading, and positive evaluation of our work.

[R3: 2] “THE EMBEDDING IN SOCIAL SCIENCE THEORY, STATISTICAL NETWORK MODELING, EMPIRICAL NETWORK RESEARCH ARE LESS ELABORATED AND I PROVIDE A NUMBER OF SUGGESTIONS BELOW. ”

Thanks for this comment (and the follow-up comments below). Please find our detailed answer below.

[R3: 3] “I) SOCIAL SCIENCE THEORY

I THINK THE AUTHORS TOOK A GREAT EFFORT IN LINKING THEIR MATHEMATICAL MODEL TO SOCIAL SCIENCE THEORY. I THINK THIS IS AN IMPORTANT STEP. HOWEVER, I PARTLY FEEL THAT THERE ARE TOO MANY SYMBOLIC CITATIONS AND TOO LITTLE ENGAGEMENT WITH THE ACTUAL ARGUMENTS IN THE PAPERS CITED. I HAVE A FEW QUESTIONS AND SUGGESTIONS ON HOW TO IMPROVE THIS PART.”

(a) “STRUCTURAL HOLES AS INTRODUCED BY BURT IS A LOCAL CONCEPT INVOLVING ACTORS IN DISTANCE ONE FROM THE FOCAL ACTOR. BETWEENNESS CENTRALITY IS DEFINED ON ALL PATHS OF A GRAPH. THE AUTHORS ARE NOT WRONG WHEN THEY SAY THAT THE CONCEPTS ARE RELATED, BUT THEN IT UNCLEAR HOW THEIR MEASURES ARE ACTUALLY A A GOOD REPRESENTATION OF STRUCTURAL HOLES MOTIFS (SEE MY COMMENT BELOW ON THE MODEL). ”

We agree with the reviewer that this relation needs to be clarified. In the payoff function section, we firstly introduce the *clustering* of agent i as a measure of weighted closed triads that surround node i , namely

$$u_i(a_i, \mathbf{a}_{-i}) = \sum_k a_{ik} \left(\sum_l a_{il} a_{lk} \right).$$

Then, we insert it within our parametric cost function which reads as:

$$V_i(a_i, \mathbf{a}_{-i}, P_i) = \alpha_i \cdot t_i(a_i, \mathbf{a}_{-i}, \delta_i) + \beta_i \cdot u_i(a_i, \mathbf{a}_{-i}) - \gamma_i \cdot c_i(a_i).$$

At this point, we discuss the role of the parameter β_i : according to the previous formula, positive values of β_i are symptomatic of an interest of agent i for closed triads, which can also be viewed as redundant ties. Conversely, negative values of β_i should be interpreted as if closed triads were acting as a cost to agent i . As we state in the manuscript:

“Drawing inspiration from [4], this enables us to measure the absence of direct brokerage opportunities and to model a number of contexts in which agents prefer ties with unconnected others, as in Burt’s theory of structural holes [5]. Albeit this cost does not correspond to the original constraint measure constructed by Burt, it preserves the underlying intuition that agents are more constrained by their network if they have many

redundant contacts."

Even if they were not considering weighted and directed networks, Burger and Buskens [4] effectively used the number of closed triads as a proxy for the Burt's network constraint, "as it comprises the notions that: (1) it is beneficial to add ties as long as these ties are non-redundant; (2) sharing one closed triad is still better than sharing more closed triads; and (3) brokerage opportunities are derived from direct contacts and not from indirect contacts. Buskens and Van de Rijt [7] show that these are the crucial properties of the utility function for predicting which network will emerge in a dynamic context", as reported in [4]. Furthermore, we emphasize that the notion of structural holes [5] introduced by Burt, is tightly related to the betweenness centrality measure, as shown by the author himself [6] and by Buechel and Buskens in the context of strategic network formation model [3].

Finally, according to the follow-up comment [R3:5(c)], we believe that the misunderstanding partially comes from the fact that we do not consider that brokers might want to destroy others' brokerage opportunities. This is certainly an interesting direction, and we further discuss on that in the answer to [R3:5(c)].

- (b) "HEIDER'S THEORY IS NOT RELATED TO THE "PSYCHOLOGICAL PRINCIPLE OF COHESION AND SUPPORT". COHESION IS CORRECTLY IDENTIFIED AS AN OUTCOME, BUT HEIDER'S ARGUMENT ABOUT WHY TRANSITIVE STRUCTURES EMERGE BUILDS UPON FESTINGER'S COGNITIVE DISSONANCE THEORY. IT ARGUES THAT IMBALANCED SITUATION MAY CAUSE STRESS FOR INDIVIDUALS AND THEREFORE THE NEED TO BALANCE THEM OUT (BY FORMING BALANCED TRIADS OR DISSOLVING IMBALANCED TRIADS). BESIDES THE FACT THAT BOTH LITERATURE STRANDS EXPLAIN THE FORMATION OF TRIADS, SOCIAL SUPPORT AND BALANCE THEORY ARE VERY DIFFERENT. THERE ARE A NUMBER OF ADDITIONAL EXPLANATIONS FOR TRIADIC CLOSURE THAT COULD EQUALLY BE CONSIDERED, FOR EXAMPLE, TRANSITIVITY AS A RESULT OF HOMOPHILY, SOCIAL FOCI OR SPATIAL STRUCTURES."

We thank the reviewer for this comment and we agree that we have not been precise on this. Thus, in the revised version of the manuscript, we sharpen the sentence highlighted by the reviewer. It now reads as:

"According to Coleman [11], triangulated structures provide cohesive support to the agents. Davis [12] also showed empirically that transitivity, often termed network (or triadic) closure or clustering [9, 30], is a prevalent effect in many human social networks as the result of social selection based, e.g. on homophily [24]."

- (c) "ARGUE WHY IT WOULD BE REASONABLE THAT TWITTER NETWORKS (GIVEN AS AN EXAMPLE) ARE THE CONSEQUENCE OF A STRATEGIC NETWORK FORMATION. WHAT ABOUT ALTERNATIVE AND POTENTIALLY UNOBSERVED EXPLANATIONS SUCH AS ALGORITHMS, COGNITIVE BIASES, OTHER TYPES OF RANDOMNESS? THIS QUESTION RELATES TO THE LACK OF AN ERROR TERM IN THE MODEL DISCUSSED BELOW. "

Certainly there is evidence that Twitter (or Instagram) are (or at least became) partially strategic networks. For instance, there exists a number of companies whose business is to help boosting one's Twitter or Instagram profiles by strategically retweeting or reposting. Undoubtedly, several business profiles are aimed at (strategically) increasing their audience to enhance marketing opportunities [34]. However, as pointed out by the reviewer, there could potentially be other unobserved explanations underlying the strategic formation of networks as, for instance, Twitter. Such a discrepancy becomes tangible when dealing with real world data. In order to account for this, we revised the behavior estimation method allowing for an error term. We discuss this issue in more depth in reply to other more specific questions below (see [R3:4(a),(d)]).

- (d) "I WOULD SUGGEST TO SIMILARLY DISCUSS SOME SOCIAL SCIENCE THEORY ON RECIPROCITY AND CYCLE STRUCTURES IN NETWORKS (SEE MY SUGGESTION ON THE MODEL BELOW)."

We thank the reviewer for this constructive comment. We defer the detailed answer to this point to a later comment below (see [R3:5(b)]). In short, we introduced, discussed and analyzed the role of reciprocity and cyclic structures in our payoff function throughout the entire revised manuscript.

[R3: 4] “II) LINK TO INFERENTIAL STATISTICS

THE AUTHORS CLAIM THAT “WITH OUR PARAMETRIC MODEL, WE ARE ABLE TO REVERSE THE TYPICAL APPROACH IN THE STRATEGIC NETWORK FORMATION LITERATURE AND INFER INDIVIDUAL INCENTIVES FROM STABLE NETWORK ARCHITECTURES.” I WOULD CALL THIS REVERSED APPROACH INFERENTIAL STATISTICS AND I WOULD FURTHER SUGGEST TO MORE THOROUGHLY REVIEW THE LITERATURE ON INFERENTIAL NETWORK METHODS (SEE, E.G., THE TEXTBOOK BY ROBINS, 2015). THERE IS IN FACT A FEW DECADES OF WORK ON HOW TO INFER INDIVIDUAL INCENTIVES AND MOTIVATION FROM STABLE NETWORK STRUCTURES. MOST RELEVANT MIGHT BE THE WORK ON EXPONENTIAL RANDOM GRAPH MODELS (SEE, E.G., LUSHER ET AL., 2013). GIVEN THE FACT THAT THE AUTHORS ASSUME THAT “EACH AGENT OF THE NETWORK HAS CONTROL ON THE WEIGHTS OF HER OUTGOING LINKS, WHILE SHE CANNOT CHANGE HER INCOMING LINKS” A REFERENCE TO AND COMPARISON WITH STOCHASTIC ACTOR-ORIENTED MODELS (SNIJDERS, 1996) MIGHT BE USEFUL AS WELL. ”

We gratefully thank the reviewer for this comment which pointed out a weakness in the analysis contained in the old version of our manuscript. We really appreciated all the suggestions which have been the starting point of an exciting journey through the world of inferential network methods. We agree with the reviewer that exponential random graph models, and especially stochastic actor oriented models have been the most relevant. As a matter of fact, we decided not only to review this strand of literature, but also to establish a connection between our strategic network formation model and these other approaches through a re-interpretation of our payoff function. We dedicated the section “Socio-theoretical interpretation” in the revised manuscript to build this parallelism. Furthermore, elaborating this connection allowed us to reshape our behavior estimation method in a statistically rigorous framework.

With regards to the latter, (see more detailed comments below), we introduced an error term in our payoff function and we define a Least Square method which allows to: (i) identify the individual behavior which minimizes the sum of squares and (ii) properly build confidence intervals of these estimates. The whole section “Inference of Behavior for Complex Networks” has been reviewed, as well as the corresponding part in the Methods section and in the SI.

- (a) “WHAT IS THE QUALITY CRITERION OF THE ACHIEVED EMPIRICAL RESULTS IN TERMS OF DEVIATION FROM AN OPTIMAL NASH EQUILIBRIUM? PROVIDING AND CALCULATING SUCH A CRITERION WOULD BE IMPORTANT AND SIMILAR TO, FOR EXAMPLE, EXPLAINED VARIANCE CRITERIA IN BASIC REGRESSION MODELS. IT COULD BE USED FOR MODEL EVALUATION AND COMPARISON. ”

Unfortunately, it is not entirely clear to us the meaning of “optimal Nash equilibrium” in the reviewer’s comment. Thus, we try to give answer to two different interpretations. In a game-theoretical context, *optimality* of Nash equilibrium might refer to its efficiency, thus as ratio between the social welfare, e.g., the sum of the agents’ payoffs, at the Nash equilibrium and the maximum social welfare achievable. This is definitely an interesting direction, however the analysis of efficient Nash equilibria remains outside the scope of our analysis. As an alternative interpretation, we assume the optimality criterion suggested by the reviewer is linked to a measure of distance with respect to the Nash equilibrium conditions. We agree with the reviewer that not enough attention was dedicated to this concept. In the revised version of the paper, after having discussed a reformulation of the payoff function and having introduced an alternative description of the individual parameter space Θ (see our answers to the game-theoretical model [R3:5]), we define the Nash equilibrium distance function which reads as follows:

$$d_i(\theta_i) := \left(\int_{\mathcal{A}} (\max \{0, e_i(a_i, \theta_i)\})^2 da_i \right)^{1/2},$$

where $e_i(a_i, \theta_i) := V_i(a_i, a_{-i}^*, \theta_i) - V_i(a_i^*, a_{-i}^*, \theta_i)$ is the error (residual) function, measuring the deviation of the payoff function with respect to the value at the strategy a_i^* . In other words, the

distance function potentially reaches its minimum value 0 for the values $\theta_i \in \Theta$ such that

$$V_i(a_i, a_{-i}^*, \theta_i) \leq V_i(a_i^*, a_{-i}^*, \theta_i), \quad \forall a_i \in \mathcal{A},$$

thus when the Nash equilibrium condition is satisfied. Conversely, the distance function takes strictly positive values whenever there exists positive violations of the Nash equilibrium conditions, i.e., when there exists $a_i \in \mathcal{A}$ such that $e_i(a_i, \theta_i) > 0$.

As described in the revised version of the manuscript (see the section “Inference of Behavior for Complex Networks”, Methods, and the SI), the minimizer(s) of the distance function is (are) regarded as the best estimates of the individual behavior. Moreover, we use the minimum value of the distance function to derive confidence intervals on the estimates, as suggested by the reviewer. We will return to this point below.

- (b) “IS THE MAIN GOAL OF THE ML APPROACH IN FACT TO “RECOGNIZING AND CLUSTERING SIMILAR INDIVIDUAL BEHAVIORS”? THIS IS A RATHER UNTYPICAL CASE STUDY, AND IF THIS IS THE MAIN POINT, THIS SHOULD BE HIGHLIGHTED MORE CLEARLY. THE CLUSTERING APPROACH REMINDS ME OF LATENT STATISTICAL MODELS AND AGAIN IT WOULD BE HELPFUL TO PROVIDE A BRIEF COMPARISON. ”

We thank the reviewer for carefully reading. In point of fact, the goal of our behavior estimation method does not consist in “recognizing and clustering similar individual behaviors”. Rather, our aim is to precisely quantify the individual behaviors. Observing similar behavior among individuals should be regarded as a possible validation of our model, yet it falls outside the main scope of the analysis, which remains the statistical estimation of the single individual’s behavior. Thus, we removed the sentence as it could divert the attention from the main scope.

- (c) “IN THE HETEROGENEOUS MODEL, THE GOAL IS NOT ANYMORE TO PROVE UNDER WHAT PARAMETERS A (P)NE OPTIMUM CAN BE FOUND FOR A GIVEN NETWORK, BUT UNDER WHICH NODE PARAMETERS THE MODEL IS GETTING AS CLOSE AS POSSIBLE TO AN NE (SEE THE COMMENT ABOVE ON A LACK OF A QUALITY CRITERION). IT IS GREAT THAT THE ML ESTIMATION WORKS. BUT IN ANALOGY TO INFERENTIAL STATISTICS I WOULD STRONGLY SUGGEST TO PROVIDE SOME MEASURE OF CONFIDENCE (SIMILAR TO STANDARD ERRORS/ CONFIDENCE INTERVALS). A PRACTICAL WAY HOW TO APPROACH THIS PROBLEM COULD BE TO RE-ESTIMATE PARAMETERS WITH A NETWORK THAT IS SLIGHTLY PERTURBED. IF NODE ESTIMATES ARE STABLE, THIS INCREASES THE CONFIDENCE IN THE SUBSEQUENT NODAL CLUSTERING. ”

Again, we greatly thank the reviewer not only for pointing out this limitation but also for providing a constructive solution. Even though we explored the suggested path focused on perturbation analysis, we noted that perturbation studies are not scalable to large networks, since the number of perturbations explodes. Hence, we decided to pursue a more standard approach in statistical inference. The method is now extensively described in the revised version of the manuscript (see the Methods section and the SI), nonetheless we would like to summarize here our ideas.

Firstly, as emphasized by the reviewer, in the behavior estimation method (previously belonging to the “heterogeneous model” section) the goal is to find the node parameters such that the model is getting as close as possible to a Nash equilibrium. Indeed, we revised the corresponding section in order to convey this objective in a rigorous framework by (i) introducing an error term, (ii) defining a distance function, and (iii) developing an ad hoc statistical inference method (see also [R3:4(a,d)] for the first two points).

As previously mentioned, the “optimal” parameters are then defined as the minimizers of the distance function, which is built on to an error function derived from the Nash equilibrium conditions. Solving this optimization problem is not an easy task, as its objective function involves a n -dimensional integral of a non-smooth function. Equivalently, it corresponds to an integral of a smooth function over a subset of the hypercube $[0 - 1]^n$. However, we established convexity and differentiability, and

thus we are able to use very robust and scalable methods to solve the optimization problem; e.g., projected gradient method. Approximating the integral with a quadrature formula allows to recast our problem to a Ordinary Least Square problem. Thanks to this analogy, it has been possible to build our statistical inference method up to the definition of the confidence intervals for the estimates. It is worth to emphasize the only difference with the standard analysis of an Ordinary Least Square problem, i.e., in our set up the error terms are always non-negative. This difference affects our analysis in two ways: (i) the estimates must be corrected (they are biased) and (ii) the computation of the confidence intervals requires a simulation-based estimator of the distribution of the error terms. A rigorous description of the difference is now provided in the SI.

- (d) “WHY IS THERE NO ERROR TERM IN THE STATISTICAL MODEL? IT SEEMS TO ME THAT AT LEAST SINCE THE WORK OF ARROW, LUCE AND OTHERS IN THE 60’S RANDOM UTILITY MAXIMIZATION MODELS ARE CONSIDERED A STANDARD IN STRATEGIC CHOICE MODELS; AT LEAST WHEN EMPIRICAL DATA IS FITTED. I GUESS THE REASONS ARE ANALYTICAL (WHICH IS COMPLETELY FINE) BUT IT SHOULD BE EXPLAINED WHY THEY OPTED FOR A NO-RANDOM UTILITY MODEL?”

As previously discussed, the reviewer’s comments evidenced the need of an error term, typically considered when empirical data is fitted to a strategic model. Such an error term, introduced in this revised version, is meant to account as a possible explanation to otherwise irrational violations of the Nash equilibrium conditions. Concretely, it is defined as follows:

$$e_i(a_i, \theta_i) := V_i(a_i, a_{-i}^*, \theta_i) - V_i(a_i^*, a_{-i}^*, \theta_i).$$

Introducing such an error term has been relevant, for instance, in the analysis of the Australian bank data-set, as well as of the new Medici data-set (see [R2:3(c), (d)]). In the Australian bank data-set, in fact, 8 out of 11 agents have proven to be irrational (according to our model specifications). Similarly, the case of the Medici family required an error term.

[R3:5] “III) GAME-THEORETICAL MODEL”

- (a) “THE AUTHORS CONSIDER THAT TIES ARE COSTLY (GAMMA PARAMETER). BUT COULD THE MODEL ALSO CONSIDER THAT THE CHANGE OF TIES COULD BE COSTLY AS WELL, NOT JUST THEIR MAINTENANCE? I UNDERSTAND THAT THIS MIGHT LEAD TO SOME MATHEMATICAL COMPLICATIONS, BUT MAYBE THE REASONS FOR TIE COSTS COULD AT LEAST BE DISCUSSED BRIEFLY. ”

We thank the reviewer for this comment and interesting observation. Although modelling the cost of changing ties differently from the cost of maintaining ties would make the model more realistic, we think that introducing new parameters to our model should be avoided unless strictly necessary. Generally, we believe that an over-parametrized model might loose its predictive power, despite being closer to reality. Our model is the result of a combination of several ideas belonging to the stream of literature on strategic network formation. Its power lies in being able to be matched with previous results, while remaining tractable for a number of extensions, e.g., comparison with random network models (see our answer below) and analysis of real-world networks.

We also would like to emphasize that we derived (unpublished and not yet submitted) analytical results for a wider class of payoff functions, where we allow for a quadratic function to model the cost of maintaining ties. A similar idea was used, for instance, in [4]. In the interest of a simple, yet realistic and flexible model, though, we decided not to include it in this paper. Similarly, we believe that, should such a variation of the payoff function be strongly supported by socio-economical aspects that are otherwise neglected, it could be included without leading to dramatic mathematical complications.

To conclude, as suggested by the reviewer, we discussed the tie cost in the manuscript “Alternatively, as a minor variation of the model, a quadratic cost function as in [4] can be used to model the fact agents have to divide their attention over all their relationships”,

as well as we added it to our future work list:

“We emphasize that our model can be adapted to different descriptions of the payoff function, e.g., considering an extra cost for changing ties, or other individual incentives such as eigenvector centrality, or constraining competitors’ brokerage [...]” .

- (b) “I THINK IT SHOULD BE CLARIFIED EARLIER ON THAT THE EXTENDED INDEGREE MEASURE (“INFLUENCE”) OF FUNCTION $f()$ AMONG OTHERS ALSO INCLUDES RECIPROCAL STRUCTURES (IF YOU FOLLOW ME, I FOLLOW YOU) AND THREE-CYCLE STRUCTURES. BOTH STRUCTURES ARE CENTRAL MOTIFS IN NETWORK MODELS AND SHOULD THUS ALSO APPEAR IN THE INTRODUCTORY SECTION. THERE IS, FOR EXAMPLE, A BROAD BODY OF LITERATURE IN HOW FAR CYCLES ARE ATYPICAL STRUCTURES IN HIERARCHICAL NETWORKS (E.G., DAVIS 1970). THE CYCLE STRUCTURE IS IN THE LITERATURE OFTEN CONTRASTED TO THE TRANSITIVITY MOTIF DISCUSSED NEXT. INTRODUCING RECIPROCITY EXPLICITLY WILL MAKE IT INTUITIVELY CLEAR WHY, FOR EXAMPLE, THE EMPTY GRAPH IS ONLY A PNE WHEN THE COSTS OF TIE FORMATION ARE HIGHER THAN THE BENEFITS OF PAIRWISE (= RECIPROCAL) COORDINATION.”

We thank the reviewer for this observation. We agree that it should be clarified in the paper. To overcome this limitation, we immediately emphasized it

“This measure can also be viewed as an approximated Katz centrality. In the original definition, Katz [17] considers paths of all lengths, yet in real-world social networks agents have limited information on the network topology (one can think of LinkedIn’s 3rd degree of separation). Compared to that, our definition is locally-assessable, i.e., it does not require complete information of the entire network, yet it includes most important social networks patterns, such as diads and triads [30].”,

as well as we dedicated an entire new section (in the revised manuscript) called “Socio-theoretical interpretation” where we highlight the presence of reciprocal and cyclic structures (sketched now in Fig. 2) embedded in the extended indegree measure.

“If we focus on the extended indegree centrality measure $t_i(a_i, \mathbf{a}_{-i}, \delta_i)$, by isolating agent i ’s contribution we obtain the following expression

$$t_i(a_i, \mathbf{a}_{-i}, \delta_i) = f_i(\mathbf{a}_{-i}, \delta_i) + \delta_i \underbrace{\left(\sum_{k \neq i} a_{ik} a_{ki} \right)}_{rec(a_i, \mathbf{a}_{-i})} + \delta_i^2 \left(\underbrace{\left(\sum_{k \neq i} a_{ik} a_{ki} \right)}_{rec(a_i, \mathbf{a}_{-i})} \underbrace{\sum_{m \neq i} a_{mi}}_{indeg(\mathbf{a}_{-i})} + \underbrace{\left(\sum_{l \neq i} a_{il} \sum_{k \neq i, l} a_{lk} a_{ki} \right)}_{cycles(a_i, \mathbf{a}_{-i})} \right).$$

In other words, the extended indegree centrality measure includes, among others, reciprocal structures (denoted as $rec(a_i, \mathbf{a}_{-i})$) and three-cycle structures, denoted as $cycles(a_i, \mathbf{a}_{-i})$.” Focusing on this aspect also allowed us to emphasize the similarities with Stochastic Actor-Oriented Models. Furthermore, it helped in interpreting our results from a sociological point of view and gave support to our findings. For instance, in the revised version we highlight the lack of cyclic structures in the Australian bank data set we analyzed (see [R2:3(c)]). Such an observation finds its validation in the work of Davis [12], as mentioned by the reviewer.

- (c) “IT IS ARGUED THAT THE BETA PARAMETER CAN BE NEGATIVE AND THEN BE IN LINE WITH BURT’S THEORY ON STRUCTURAL HOLES. IT WOULD THEN BE LESS COMMON THAT TWO PATHS a_{il} a_{lk} ARE CLOSED BY A DIRECT CONNECTION a_{ik} . BUT ISN’T IT INDIVIDUAL l WHO IS ACTUALLY THE BROKER IN THIS MODEL AND THUS THE ONE WHO HAS AN INCREASED UTILITY BY BRIDGING A STRUCTURAL HOLE? WHY WOULD i HAVE AN INCENTIVE NOT TO DESTROY l ’S BROKERAGE POSITION? CAN YOU CLARIFY HOW THIS FITS TO YOUR ACTOR-CONTROL APPROACH?”

We agree with the reviewer that l has a brokerage position in this case. However, we assume agent i is only interested in her/his brokerage opportunities. In this example, agent i would not see an advantage in connecting to agent k as this would create redundant connections. We do not assume agent i competes with agent l to prevent l 's structural advantage. Of course, that could be an interesting direction for future investigation and we thank the reviewer for pointing this out stimulating the following change in the conclusions:

"We emphasize that our model can be adapted to different descriptions of the payoff function, e.g., considering an extra cost for changing ties, or other individual incentives such as eigenvector centrality, or constraining competitors' brokerage".

[R3:6] "IV) SUGGESTIONS ABOUT THE FRAMING THE FOLLOWING TWO COMMENTS RELATE TO THE FRAMING OF THE STUDY. THEY ARE MERELY SUGGESTIONS AND NOT REQUESTS IN TERMS OF THE FORTHCOMING R&R."

- (a) "THE RESULTING NETWORKS ARE HIGHLY STYLIZED (E.G., EMPTY, COMPLETE, STAR GRAPH). THERE IS A STRAND IN THE NETWORK SCIENCE LITERATURE (HIGHLY CITED, LESS USEFUL FOR PRACTICAL PURPOSES) IN WHICH SIMILARLY STYLIZED NETWORK MODELS HAVE BEEN PROPOSED AND FITTED TO DATA. EXAMPLES ARE PREFERENTIAL ATTACHMENT MODELS, STOCHASTIC BLOCKMODELS, SMALL WORLD MODELS. IT MIGHT BE AN IDEA TO CONNECT TO THE ANALYSIS TO SUCH MODELS AND EXPLAIN WHETHER AND IN HOW MUCH THEY DEVIATE FROM PERFECT NE OUTCOMES. "

We thank the reviewer for her/his comment. Even though that was not requested, we considered it a significant extension of our work, also suggested by Reviewer #1 (See [R1:3,7]). Hence, we introduced a new section called "Random Networks" where we use our behavior estimation method to give a sociological and strategic interpretation of the probabilistic rules behind two well known examples of random networks, namely the Preferential Attachment and the Small-world models.

- (b) "ANOTHER LITERATURE THAT MIGHT BE WORTH EXPLORING TO BUILD UPON IS THE WORK ON MODEL DEGENERACY (AND THE CONDITIONS OF DEGENERACY) FOR EXPONENTIAL RANDOM GRAPH MODELS."

We thank the reviewer for this constructive suggestion which we temporarily leave for successive investigations.

We finally thank the reviewer again for all her/his constructive comments which prompted a lot of changes in the paper and led to an improved manuscript.

References

- [1] Abhijit Banerjee, Esther Duflo, and Richard Hornbeck. Bundling health insurance and microfinance in india: There cannot be adverse selection if there is no demand. *American Economic Review*, 104(5):291–97, 2014.
- [2] Albert-László Barabási and Réka Albert. Emergence of scaling in random networks. *science*, 286(5439):509–512, 1999.
- [3] Berno Buechel. Network formation with closeness incentives. *Networks, Topology and Dynamics. Springer Lecture Notes in Economic and Mathematical Systems*, 613:95–109, 2008.
- [4] Martijn J. Burger and Vincent Buskens. Social context and network formation: An experimental study. *Social Networks*, 31(1):63–75, 2009.

- [5] Ronald S Burt. Structural hole. *Harvard Business School Press, Cambridge, MA*, 1992.
- [6] Ronald S Burt. Bridge decay. *Social networks*, 24(4):333–363, 2002.
- [7] Vincent Buskens, Rense Corten, and Jeroen Weesie. Consent or conflict: Coevolution of coordination and networks. *Journal of Peace Research*, 45(2):205–222, 2008.
- [8] Guido Caldarelli. *Large scale structure and dynamics of complex networks: from information technology to finance and natural science*, volume 2. World Scientific, 2007.
- [9] Dorwin Cartwright and Frank Harary. Structural balance: a generalization of heider’s theory. *Psychological review*, 63(5):277, 1956.
- [10] Claudio Castellano, Santo Fortunato, and Vittorio Loreto. Statistical physics of social dynamics. *Reviews of modern physics*, 81(2):591, 2009.
- [11] James Coleman. Foundations of social theory. *Cambridge, MA: Belknap*, 1990.
- [12] James A Davis. Clustering and hierarchy in interpersonal relations: Testing two graph theoretical models on 742 sociomatrices. *American Sociological Review*, pages 843–851, 1970.
- [13] Paul Erdős and Alfréd Rényi. On random graphs, i. *Publicationes Mathematicae (Debrecen)*, 6:290–297, 1959.
- [14] Paul Erdős and Alfréd Rényi. On the evolution of random graphs. *Publ. Math. Inst. Hung. Acad. Sci*, 5(1):17–60, 1960.
- [15] Sébastien Grauwain, Eric Bertin, Rémi Lemoy, and Pablo Jensen. Competition between collective and individual dynamics. *Proceedings of the National Academy of Sciences*, 106(49):20622–20626, 2009.
- [16] Jaime Iranzo, Javier M Buldú, and Jacobo Aguirre. Competition among networks highlights the power of the weak. *Nature communications*, 7:13273, 2016.
- [17] Leo Katz. A new status index derived from sociometric analysis. *Psychometrika*, 18(1):39–43, 1953.
- [18] Dale V Kent. *The rise of the Medici: Faction in Florence, 1426-1434*. Oxford University Press, USA, 1978.
- [19] Michael D Koenig and Stefano Battiston. From graph theory to models of economic networks. a tutorial. *Networks, topology and dynamics*, 613:23–63, 2009.
- [20] Amy N Langville and Carl D Meyer. *Google’s PageRank and beyond: The science of search engine rankings*. Princeton University Press, 2011.
- [21] Vito Latora, Vincenzo Nicosia, and Giovanni Russo. *Complex networks: principles, methods and applications*. Cambridge University Press, 2017.
- [22] Dean Lusher, Johan Koskinen, and Garry Robins. *Exponential random graph models for social networks: Theory, methods, and applications*. Cambridge University Press, 2013.
- [23] Andreu Mas-Colell, Michael Dennis Whinston, Jerry R Green, et al. *Microeconomic theory*, volume 1. Oxford university press New York, 1995.
- [24] Miller McPherson, Lynn Smith-Lovin, and James M Cook. Birds of a feather: Homophily in social networks. *Annual review of sociology*, 27(1):415–444, 2001.
- [25] Stanley Milgram. The small world problem. *Psychology today*, 2(1):60–67, 1967.

- [26] Mark Newman. *Networks: an introduction*. Oxford university press, 2010.
- [27] John F Padgett and Christopher K Ansell. Robust action and the rise of the medici, 1400-1434. *American journal of sociology*, 98(6):1259–1319, 1993.
- [28] Romualdo Pastor-Satorras and Alessandro Vespignani. Epidemic spreading in scale-free networks. *Physical review letters*, 86(14):3200, 2001.
- [29] Philippa Pattison, Stanley Wasserman, Garry Robins, and Alaina Michaelson Kanfer. Statistical evaluation of algebraic constraints for social networks. *Journal of mathematical psychology*, 44(4):536–568, 2000.
- [30] Garry Robins, Pip Pattison, Yuval Kalish, and Dean Lusher. An introduction to exponential random graph (p^*) models for social networks. *Social networks*, 29(2):173–191, 2007.
- [31] Tom AB Snijders. Stochastic actor-oriented models for network change. *Journal of mathematical sociology*, 21(1-2):149–172, 1996.
- [32] Tom AB Snijders. The statistical evaluation of social network dynamics. *Sociological methodology*, 31(1):361–395, 2001.
- [33] Tom AB Snijders, Gerhard G Van de Bunt, and Christian EG Steglich. Introduction to stochastic actor-based models for network dynamics. *Social networks*, 32(1):44–60, 2010.
- [34] Tracy L Tuten and Michael R Solomon. *Social media marketing*. Sage, 2017.
- [35] Duncan J Watts and Steven H Strogatz. Collective dynamics of ‘small-world’ networks. *nature*, 393(6684):440, 1998.

Reviewers' Comments:

Reviewer #1:

Remarks to the Author:

The authors have addressed my comments and requirements, and I believe that the paper has increased substantially in impact and applicability. For these reasons I recommend its publication in Nature Communications.

Few typos to amend:

Pag. 1: Strives for centrality >> strive for centrality

Pags. 9 and 10: rewiring probability >> rewiring probability

In Fig. 10, please include the color-code to facilitate its interpretation.

Reviewer #2:

Remarks to the Author:

The authors have made extensive revisions of the manuscript to respond to numerous reviewers' comments. While I could imagine discussing further a number of finer issues, I find the current version ready for publication.

Reviewer #3:

Remarks to the Author:

I think the authors did an exceptionally careful and thorough revision. All my concerns with regards to the earlier version have been sufficiently addressed. I recommend the article for publication.

Statement of Revision

Comments by Reviewer # 1

[R1: 1] “ THE AUTHORS HAVE ADDRESSED MY COMMENTS AND REQUIREMENTS, AND I BELIEVE THAT THE PAPER HAS INCREASED SUBSTANTIALLY IN IMPACT AND APPLICABILITY. FOR THESE REASONS I RECOMMEND ITS PUBLICATION IN NATURE COMMUNICATIONS.

FEW TYPOS TO AMEND:

PAG. 1: STRIVES FOR CENTRALITY → STRIVE FOR CENTRALITY

PAGS. 9 AND 10: REWRIRING PROBABILITY → REWIRING PROBABILITY

IN FIG. 10, PLEASE INCLUDE THE COLOR-CODE TO FACILITATE ITS INTERPRETATION. ”

We thank the reviewer for the positive evaluation of our work and for reporting the typos. We fixed them in the latest version. We also thank the reviewer again for all her/his constructive comments which prompted a lot of changes in the paper and led to an improved manuscript.

Comments by Reviewer # 2

[R2: 1] “THE AUTHORS HAVE MADE EXTENSIVE REVISIONS OF THE MANUSCRIPT TO RESPOND TO NUMEROUS REVIEWERS’ COMMENTS. WHILE I COULD IMAGINE DISCUSSING FURTHER A NUMBER OF FINER ISSUES, I FIND THE CURRENT VERSION READY FOR PUBLICATION. ”

We thank the reviewer for the positive evaluation and for all her/his constructive comments which led to an improved manuscript.

Comments by Reviewer # 3

[R3: 1] “I THINK THE AUTHORS DID AN EXCEPTIONALLY CAREFUL AND THOROUGH REVISION. ALL MY CONCERNS WITH REGARDS TO THE EARLIER VERSION HAVE BEEN SUFFICIENTLY ADDRESSED. I RECOMMEND THE ARTICLE FOR PUBLICATION.”

We thank the reviewer again for all her/his constructive comments which prompted a lot of changes in the paper and led to an improved manuscript.